# Robust 2D layered MXene matrix–boron carbide hybrid films for neutron radiation shielding

Ju-Hyoung Han[1,6], Shi-Hyun Seok[1,6], Young Ho Jin[1], Jaeeun Park[1], Yunju Lee [2], Haeng Un Yeo[1], Jong-Ho Back[3], Yeoseon Sim[1], Yujin Chae[1], Jaewon Wang [1], Geum-Yoon Oh[4], Wonjoo Lee[3], Sung Hyun Park [4], In-Cheol Bang[2], Ji Hyun Kim [2] & Soon-Yong Kwon [1,5] ✉

Large-scale fabrication of neutron-shielding films with flexible or complex shapes is challenging. Uniform and high boron carbide ($B_4C$) filler loads with sufficient workability are needed to achieve good neutron-absorption capacity. Here, we show that a two-dimensional (2D) $Ti_3C_2T_x$ MXene hybrid film with homogeneously distributed $B_4C$ particles exhibits high mechanical flexibility and anomalous neutron-shielding properties. Layered and solution-processable 2D $Ti_3C_2T_x$ MXene flakes serve as an ideal robust and flexible matrix for high-content $B_4C$ fillers (60 wt.%). In addition, the preparation of a scalable neutron shielding MXene/$B_4C$ hybrid paint is demonstrated. This composite can be directly integrated with various large-scale surfaces (e.g., stainless steel, glass, and nylon). Because of their low thickness, simple and scalable preparation method, and an absorption capacity of 39.8% for neutrons emitted from a $^{241}Am$–$^9Be$ source, the 2D $Ti_3C_2T_x$ MXene hybrid films are promising candidates for use in wearable and lightweight applications.

The rapid development of modern, powerful, and resourceful electronic devices for application in the aerospace and automotive industries, such as those for high-altitude flights, satellites, military aircraft, and urban air mobility, as well as electronic systems used in other high-radiation environments, including nuclear and medical fields, has caused some serious external radiation-induced reliability issues[1–7]. The energy particles of these external radiation randomly collide with electrical components and can cause malfunctions and failure, and as information communication technology parts become highly integrated, concerns about the effects of exposure to high-level external radiation are growing[3,8]. The absence of radiation-tolerant technology in various fields that require high reliability can not only

lead to failures in space and future mobility exploration missions but also malfunctions in social infrastructures, resulting in enormous physical and human losses.

Among the radioactive rays, neutrons have an exceptional ability to penetrate most materials and interact with atomic nuclei to form isotopes, in turn releasing more ionizing rays and triggering further neutron radiation[9–11]. Since neutrons are very harmful to human health and the environment, it is necessary to shield by appropriately using the scattering and neutron absorption mechanisms. The scattering interaction of neutrons with nuclei of similar atomic weight such as hydrogen can reduce the energy of the neutrons. However, deceleration into a thermal neutron requires many collisions (theoretically, 27

[1]Department of Materials Science and Engineering, Ulsan National Institute of Science and Technology (UNIST), Ulsan 44919, Republic of Korea. [2]Department of Nuclear Engineering, Ulsan National Institute of Science and Technology (UNIST), Ulsan 44919, Republic of Korea. [3]Center for Advanced Specialty Chemicals, Korea Research Institute of Chemical Technology (KRICT), Ulsan 44412, Republic of Korea. [4]Sustainable Technology and Wellness R&D Group, Korea Institute of Industrial Technology (KITECH), Jeju 63243, Republic of Korea. [5]Graduate School of Semiconductor Materials and Devices Engineering, Ulsan National Institute of Science and Technology (UNIST), Ulsan 44919, Republic of Korea. [6]These authors contributed equally: Ju-Hyoung Han, Shi-Hyun Seok. ✉e-mail: sykwon@unist.ac.kr

for a neutron with a kinetic energy of 2 MeV[9]. This results in thickness requirements of several centimeters for hydrogen-containing shielding materials (e.g., water, concrete, and polyethylene), which restrict the form and size of protective facilities. Instead of being scattered, neutrons can be absorbed or captured by nuclei, accompanied by the emission of charged alpha particles.

Boron carbide ($B_4C$) is widely used as a neutron-absorbing material because of its high reaction cross-section of $^{10}B$, high melting point (2763 K), and low density (2.52 g cm$^{-3}$)[12,13]. Boron naturally exists as two stable isotopes, i.e., $^{10}B$ and $^{11}B$, at a ratio of 1:4, and its neutron capture ability is mainly attributed to isotope $^{10}B$. Therefore, boron and boron compounds such as boron oxide ($B_2O_3$) and boron nitride (BN) are promising candidates for neutron radiation shielding. However, the surface density of boron, which is directly related to the neutron-shielding capabilities of boron compounds, is prominent in the order of $B_2O_3$, hexagonal BN, $B_4C$, and B[14,15]. Although B has a higher neutron-shielding ability than $B_4C$, many impurities in commercially available boron powder produce secondary gamma emissions after irradiation. In this regard, $B_4C$ is the most widely used form in neutron-shielding applications. Therefore, several composites, such as boron-containing stainless-steel[16], $B_4C$-reinforced aluminum[17], and $B_4C$-reinforced polymers[18], have been developed. Traditional boron-containing metal matrices were the materials of choice in nuclear industry; however, with the smaller modern electronic applications in the aerospace and automotive industries and other high-radiation environments, the practical application of these composites as flexible or complex shapes has been limited owing to their insufficient workability. In addition, the lack of solubility of boron in metallic matrices produces heterogeneous structures, including grain boundaries, segregation, and compounds, resulting in low boron content and uneven distribution[19–21]. Moreover, densification of the $B_4C$ monolithic body via casting and pressureless sintering is challenging because of its low plasticity, low self-diffusion coefficient, and high brittleness[22]. The homogeneous mixing of boron in a polymer matrix requires surface modification to enhance the interfacial interactions, thereby reducing processability[23]. The chemical functionalization of inorganic particles or boron molecular clusters (e.g., carborane) for blending with polymers is generally expensive and time-consuming, thus greatly limiting their commercialization[24]. The aqueous dispersion of $B_4C$ powders also requires high solid loading or an additional dispersing agent for colloidal formation[25,26]. Therefore, designing lightweight, shape-controllable, structurally stable $B_4C$ composites using a cost-effective technique and gaining detailed insight into boron distribution in the robust and flexible matrix is critical for constructing effective neutron-shielding measures.

Structures for various applications ranging from energy storage to environmental applications[27], optoelectronics[28], and electromagnetic interference (EMI) shielding[29] have been developed using the tunable properties of two-dimensional transition metal carbides (MXenes). 2D MXenes have recently attracted considerable attention in various fields as an ideal building block for fabricating strong and conductive films[30,31]. However, although such materials are highly promising, the assembly of 2D MXenes with guest species for tailoring functional multiscale architectures remains a challenge. Here, we report that 2D flakes of titanium carbide ($Ti_3C_2T_x$) MXene, with large flake sizes, produced by a simple and scalable solution-mixing technique without the need for any heating or casting, can be used as a robust and flexible matrix for $B_4C$ fillers with high content. With the layered 2D MXene-based assembly, high neutron-shielding ability, far superior to that of conventional $B_4C$ composites, was achieved in both vacuum-filtrated and painted hybrid films at an ultralow thickness, enabling them to effectively shield surfaces of any shape.

## Results

### Solution-based process for MXene/$B_4C$/PVA hybrid films

Figure 1 shows the strategy for a simple preparation of MXene/$B_4C$/polyvinyl alcohol (PVA) (MBP) hybrid films. As-prepared $Ti_3AlC_2$ MAX phase and chemically exfoliated $Ti_3C_2T_x$ MXene flakes with a large lateral size exhibited a high quality (Fig. 1a, Supplementary Figs. 1 and 2, Supplementary Table 1, and see details in Supplementary Note 1). Homogeneously dispersed $B_4C$ was obtained by separating small particles with sizes <300 nm through sonication and centrifugation (nano-sized $B_4C$ (n-$B_4C$)) (Fig. 1b, Supplementary Figs. 3–9, Supplementary Tables 2 and 3, and see details in Supplementary Note 2). The surface chemical state of n-$B_4C$ was modified to enhance electrostatic dispersibility, maintaining structural integrity and high crystallinity. Owing to the repulsive forces between the $Ti_3C_2T_x$ flakes and $B_4C$ particles, which resulted from their negative surface charges, a homogeneous and stable $Ti_3C_2T_x$ MXene/$B_4C$ (MB) hybrid colloid with a zeta ($\zeta$) potential of −38.2 mV was obtained. All the hybrid solutions,

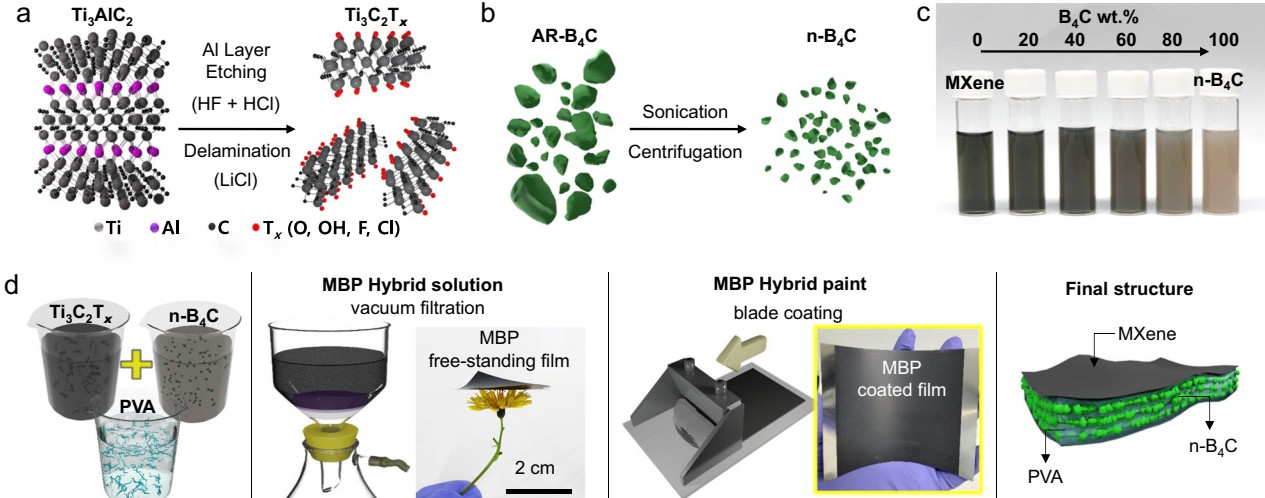

**Fig. 1 | Solution-based process for MXene composite films with incorporated $B_4C$ particles. a** Synthesis of the $Ti_3C_2T_x$ MXene through the mixed etchant (HF + HCl) and inorganic intercalant (LiCl). **b** Size selection from the as-received $B_4C$ (AR-$B_4C$) to nano-sized $B_4C$ (n-$B_4C$). **c** Stable and homogeneous dispersion of MXene/n-$B_4C$ hybrid colloid solutions with various $B_4C$ concentrations. **d** Preparation of the $Ti_3C_2T_x$/n-$B_4C$/PVA (MBP) hybrid solution and its film fabrication using vacuum-assisted filtration and blade-coating methods.

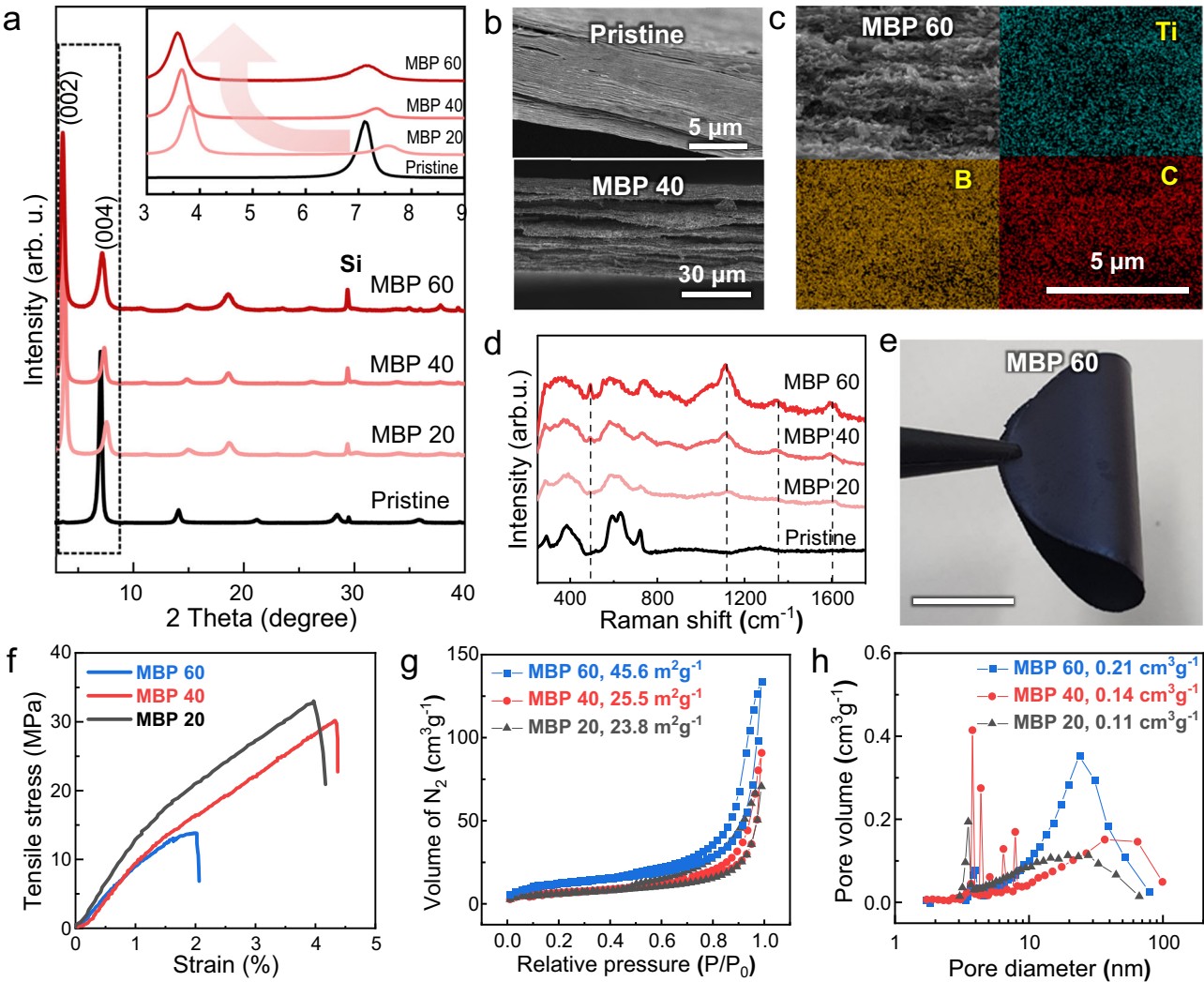

**Fig. 2 | Structural characterization of MXene/B$_4$C/PVA (MBP) hybrid films.**
**a** XRD comparison of MBP hybrid films. An enlarged pattern in the range of 2°–9° is shown in the inset. **b** Cross-sectional SEM images of the pristine Ti$_3$C$_2$T$_x$ and MBP 40 hybrid film. **c** Cross-sectional SEM image and corresponding EDS mapping results of MBP 60. **d** Raman spectra of the MBP hybrid films. The dashed lines represent characteristic peaks of B$_4$C. **e** Photograph of the MBP 60 freestanding film showing its good flexibility (scale bar, 1 cm). **f** Tensile strain–stress curves of the MBP hybrid films. **g** N$_2$ BET analysis of the MBP hybrid films. **h** BJH pore size distributions of the MBP hybrid films. Calculated surface areas and pore volumes are included in **g** and **h**, respectively.

with weight fractions of n-B$_4$C ranging from 0 to 100 wt%, maintained stability without any aggregation or sedimentation even after 7 days (Fig. 1c) and the importance of the homogeneity of the hybrid solution was further confirmed using vacuum-filtrated films (Supplementary Figs. 10 and 11 and see details in Supplementary Note 3). PVA as a binder for the Ti$_3$C$_2$T$_x$ interlayer was subsequently added to the MB hybrid solution or hybrid paint (Fig. 1d). Finally, MBP composite films were successfully fabricated using vacuum-assisted filtration or blade-coating method.

## Fabrication and mechanical properties of freestanding MBP films

To realize the applicability towards robust and flexible skeletons incorporating neutron-absorbing materials, organic PVA was added as an interlayer binder. X-ray diffraction (XRD) patterns showed that the sharp (002) peak of Ti$_3$C$_2$T$_x$ MXene at $2\theta \approx 7°$ was sequentially shifted to lower angles with increasing PVA ratios, indicating the intercalation of PVA chains, which facilitates the ordered stacking of Ti$_3$C$_2$T$_x$ flakes along [0001] (Supplementary Fig. 12a). In the previous study, the mechanical strength and flexibility of the Ti$_3$C$_2$T$_x$/PVA composites were improved compared to those of the pure Ti$_3$C$_2$T$_x$ film, suggesting

the presence of interfacial bonding between the flakes and PVA[32]. The tensile stress–strain curves of the MXene/PVA composite films with varying PVA content exhibited that the tensile strength and failure strain of the pure MXene film were significantly increased by the introduction of PVA and then decreased with a further increase in PVA content (Supplementary Fig. 12b and c). Because too many PVA chains hinder the interfacial bonding between the MXene flakes, a PVA weight fraction of 20 wt% endowed the composite film with optimal mechanical properties; thus, it was used in the fabrication of MBP hybrid films.

A freestanding MBP composite film for neutron radiation shielding, ≈30 mm in diameter, was fabricated via vacuum-assisted filtration of the hybrid dispersion of n-B$_4$C and Ti$_3$C$_2$T$_x$ with a fixed fraction of PVA loading (i.e., 20 wt%). XRD analysis of the MBP composite film confirmed an ordered lateral structure when the weight fraction of B$_4$C in the composite film varied from ≈20 to ≈60 wt% (MBP 20–60) (Fig. 2a). The (00l) peaks of MXene in the spectra indicated that the flakes were stacked laterally with good alignment and slightly shifted to lower angles with increasing B$_4$C content (inset in Fig. 2a). The cross-sectional scanning electron microscopy (SEM) image of the composite showed a highly ordered in-plane orientation with the incorporated

$B_4C$ particles in the gaps between the MXene flakes and PVA layers, forming a sandwich-like structure (Fig. 2b and Supplementary Fig. 13). The well-packed and long-range layer-by-layer microstructure could be formed by the homogeneous incorporation of the nano-sized $B_4C$ powder between the MXene flakes with sufficiently large lateral sizes up to several micrometers, whereas small MXene flakes with sizes <1 μm could not be supported even at low weight fractions of the $B_4C$ particles. The energy dispersive spectroscopy (EDS) spectra indicated a uniform distribution of representative elements of B, C, and Ti, demonstrating that $B_4C$ particles were evenly introduced within the ternary composite films (Fig. 2c). The Raman spectrum revealed that the bands of boron carbide near ≈30 and ≈1100 $cm^{-1}$, caused by chain-icosahedral linkage and icosahedral vibrational modes, respectively, appeared in all the composites (Fig. 2d)[33]. Intensities of the peaks were increased with the fraction of $B_4C$, indicating that the particles were successfully incorporated on the surface and between the layers of MXene flakes. As-fabricated freestanding MBP composite films exhibited good flexibility, and structural stability, showing uniformly aligned structures over a wide range in cross-sectional SEM images (Fig. 2e and Supplementary Fig. 13). The EDS mapping results further revealed the presence and homogeneous distribution of $B_4C$ in the $Ti_3C_2T_x$-based composite film, indicating its ability to contain high boron concentrations (Supplementary Fig. 14).

The tensile stress–strain curves of the MBP composite films with varying $B_4C$ fractions are shown in Fig. 2f. The $Ti_3C_2T_x$/20 wt% PVA film exhibited a tensile strength of 119.38 ± 11.29 MPa and failure strain of 5.26 ± 0.36% (Supplementary Fig. 12c). However, the incorporation of $B_4C$ into the hybrid composite film significantly weakened the mechanical properties. The tensile strength gradually decreased with the increase in the $B_4C$ fraction. Nevertheless, the MBP 20 and MBP 40 composite films still maintained good mechanical properties, comparable to those of other neutron-shielding $B_4C$ composite coatings or flexible films[17,34]. When compared to a commercially available resin-based shielding material (Mirrotron, Mirrobor™), our MBP film showcased lower thickness and higher mechanical strength, suggesting a reduced mechanical degradation rate as a coating material (Supplementary Fig. 15). The mechanical properties further weakened for MBP 60, likely due to the lower alignment and packing density of MXene flakes, resulting in weak inter-flake interactions[35]. The nitrogen (77 K) adsorption/desorption isotherms were conducted to understand the mechanical properties of MBP hybrid film (Fig. 2g). The MBP hybrid films showed type IV nitrogen adsorption/desorption isotherms with distinct hysteresis loops originating from the mesopores owing to the intercalation of $B_4C$, showing a similar tendency to that of the MXene-based films reported previously[36,37]. The Brunauer–Emmett–Teller (BET) surface area increased as the fraction of $B_4C$ increased from 23.8 (20 wt%) to 45.6 (60 wt%) $m^2 g^{-1}$. Notably, hybrid films with different $B_4C$ fractions had similar adsorptive structures to mesopores, suggesting the high boron capacity of MXene. The Barrett–Joyner–Halenda (BJH) desorption pore distribution was calculated to obtain more detailed information about the pore structure. The pore volume gradually increased with the intercalation of $B_4C$ (from 0.11 (20 wt%) to 0.21 (60 wt%) $cm^3 g^{-1}$); however, the pore size distribution was in a range of ≈2–60 nm, which is smaller than macropores generated from the cavities between the as-received $B_4C$ (AR-$B_4C$) particles and MXene nanosheets (Fig. 2h and Supplementary Fig. 3b). An n-$B_4C$/PVA film was also prepared as a control sample using a similar method (Supplementary Fig. 16). The film without MXene flakes exhibited not only macroscopic pores but also separation of $B_4C$ particles and PVA matrix, with non-uniform boundaries at the interface, which is one of the major fracture mechanisms in reinforced ceramic particle composites[38].

## Fabrication and mechanical properties of painted MBP films

Our vacuum-filtrated MBP hybrid film shows its applicability as a neutron-shielding coating material with a highly aligned structure,

homogeneous distribution of $B_4C$, and structural stability; however, the fabrication method still has some drawbacks for practical application in the construction of radiation shields. Thus, as a new form of neutron-shielding coating, an easily applicable MBP hybrid paint with thickness controllability was produced by concentrating the hybrid solution. Precipitating the delaminated MXene flakes was possible using centrifugation owing to their large flake sizes, which resulted in a rapid sedimentation rate[39]. The sedimented MXene flakes were redispersed in DI water by manual shaking and sonication, resulting in an MXene paint with a high concentration of ≈20 mg $ml^{-1}$. n-$B_4C$ was vacuum-filtrated and dried to obtain a powder. Subsequently, a hybrid paint with a $B_4C$ fraction of 40 wt% was fabricated by mixing the MXene paint and the desired amount of redispersed n-$B_4C$ powder with the addition of PVA. Our ternary hybrid paint with the same mass ratio as the MBP hybrid film was easily painted on a conventional stainless-steel foil (AISI 304, 100 μm) over large areas using a blade-coating method, which may be due to the high viscosity of the paint[40,41]. The versatility of the MBP paints was demonstrated by hybrid films directly realized onto curved surfaces such as concave and convex surfaces of stainless-steel foils, as depicted in Fig. 3a. The thicknesses of the painted MBP films having large areas of ≈40 × 40 $mm^2$ were measured as a function of the number of coating processes to validate the applicability of the films on desired substrates (Fig. 3b). The thickness of the MBP film increased almost linearly with the number of coating processes, which could imply a promising neutron-shielding ability. Moreover, a uniform thickness over a large area of approximately 25 mm × 50 mm was confirmed via measurements at 10 different locations by using a digital micrometer (Fig. 3c).

The cross-sectional SEM images showed that the painted MBP film was uniformly and intimately attached to various substrates while maintaining a similar morphology to those of a freestanding film produced by vacuum filtration, which suggests a stable structure without fracture or detachment (Fig. 3d). Notably, the paint could also be painted on ultraviolet ozone (UVO)-treated glass substrates (Marienfeld, ground edge slide glass), which shows the ability of MXenes to be directly integrated on desired substrates. In particular, the paint was well painted on a hydrophilic nylon fabric membrane (GVS, neutral nylon transfer membrane) due to the hydrophilicity and large surface charge of the MXene flakes that composed the paint. The surface and cross-sectional morphologies displayed in Supplementary Fig. 17 show that the painted films possessed the same structure as the vacuum-filtered films. EDS mapping results of the surface and cross-section of the painted MBP films on various substrates (i.e., stainless steel, glass, and nylon) further confirmed the even distribution of Ti, B, and C throughout the area, indicating that n-$B_4C$ was evenly mixed in the MXene matrix (Supplementary Fig. 18). A stripping test using an adhesive tape qualitatively implied a good bonding strength between the painted MBP films and the nylon substrate. The film remained on the substrate with minimal surficial disruption after tape removal (Supplementary Fig. 19). Furthermore, it can be observed that the mechanical strength of the nylon membrane painted with an MBP film did not exhibit any remarkable changes (Supplementary Fig. 20). The mechanical strength and failure strain of the hybrid film was primarily attributed to the robust material properties of the flexible substrates that were used, which allowed the films to undergo uniform stress-induced strain. The painted hybrid film on the fabric structure exhibited superior shape variability and flexibility based on the good mechanical properties of the blade-coated MXene flakes, which are expected to be applicable as a multifunctional coating and neutron-shielding textile (Fig. 3e)[35].

Bending tests were performed to thoroughly investigate the structural stability of the painted hybrid films on a fabric substrate (Supplementary Fig. 21). Prior to measurements, the composite films were heat-treated on a hot plate at 100 °C for 1 h to assess their thermal

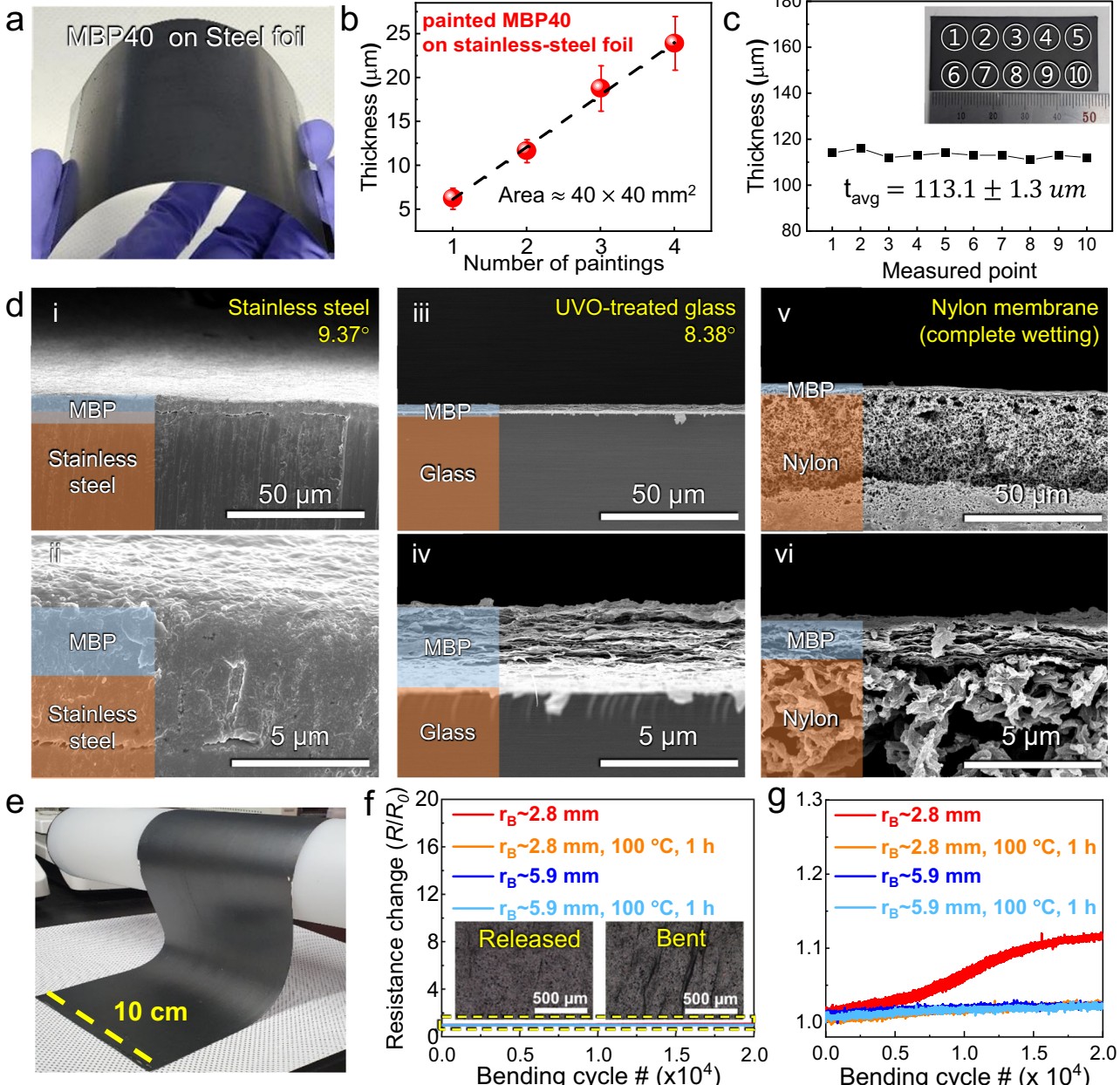

**Fig. 3 | Blade-coating of MBP hybrid paint. a** Photograph of the painted MBP hybrid film on a curved surface of stainless-steel foil. **b** Thicknesses of the painted MBP hybrid films as a function of the number of paintings. The average and standard deviation of the thicknesses of 14 different areas in each sample are represented as data points ± error bars. The error bars are 1.20, 1.30, 2.58, and 3.05 μm (from left to right). **c** Thicknesses of the films painted on stainless-steel foils (thickness ≈ 100 μm) over a large area of 25 mm × 5 mm. The inset shows an image of the painted MBP sample. **d** Cross-sectional SEM images of the MBP hybrid film on the various substrates including the stainless-steel foil (i and ii), glass (iii and iv), and nylon fabric membrane (v and vi). Contact angles of the bare substrates are included on the upper right in (i, iii, and v). **e** Photograph of the painted MBP hybrid film on the nylon membrane with a large area of 10 × 30 cm². **f** Resistance changes as a function of the bending cycle of the hybrid film of painted MBP40 on a nylon membrane at different bending radii. The insets show OM images of the released and bent MBP hybrid film/nylon membrane sample. **g** Magnified view of the yellow dashed area in (**f**).

stability. The painted hybrid film exhibited a crumpled surface without fractures or visible cracks after repeated bending tests (Supplementary Fig. 21b–d). During the tests, the film was compressed in the longitudinal direction, with varied bending radii of 5.9 and 2.8 mm using a deformation tester (i.e., CKMF-12P, CKSI). After repeated deformation, the films also demonstrated excellent flexibility with minor resistance changes (Fig. 3f). This indicates that the robust layered structure of the MXene-based film maintained intact over 20,000 cycles of bending to a small radius of curvature. Notably, the composite films heat-treated at 100 °C exhibited much-improved

flexibility, which is possibly due to the thermal rearrangement of interlayer PVA chains over the glass transition temperature of ≈80 °C. Our results imply that we have more room for improvement in the thermal stability of MBP hybrid films by introducing appropriate high-temperature resistive polymers (e.g., polyimide[42]).

**Neutron-shielding ability of freestanding and painted MBP films**
The neutron radiation shielding ability of the MBP hybrid films with varying $B_4C$ content and thickness was estimated using thermal neutron attenuation tests. The neutron permeability of the freestanding

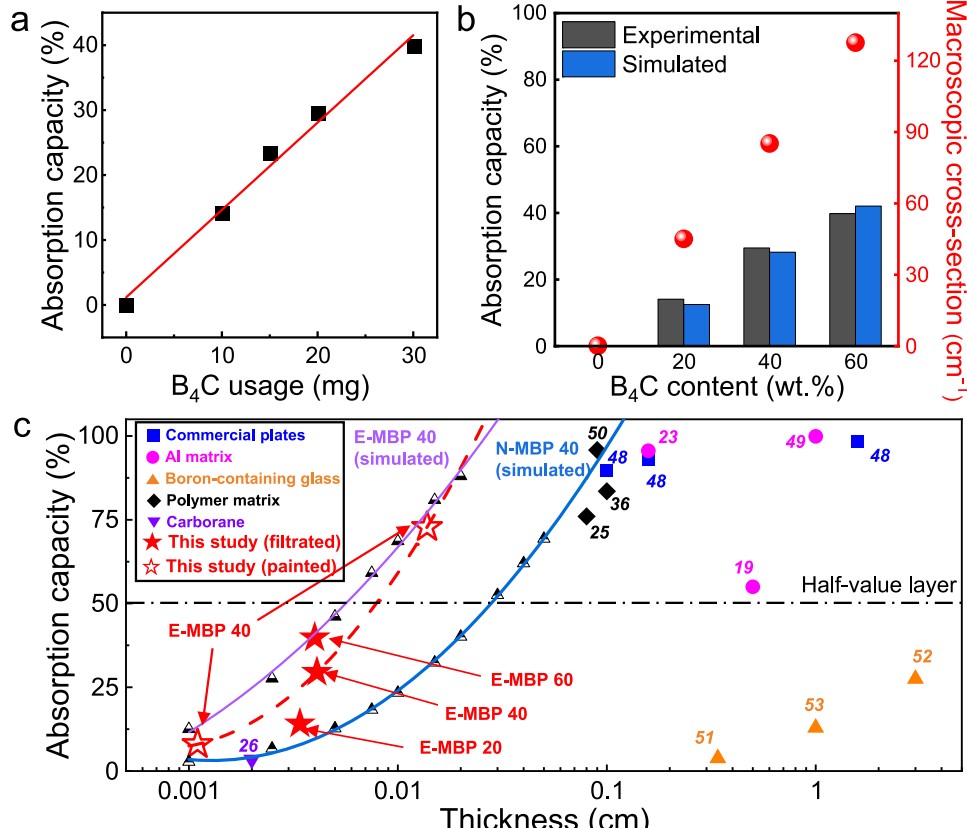

**Fig. 4 | Neutron-shielding performances of MBP hybrid films. a** Neutron absorption capacity of MBP hybrid films with different $B_4C$ usages. **b** Neutron absorption capacity and calculated macroscopic absorption cross-section of the hybrid films with varying $B_4C$ content. The analysis involved a comparison between the experimental and simulated results. **c** Neutron absorption capacity vs. thickness of boron-based composites. The dashed lines in (**c**) represent the fitted data. Detailed neutron-shielding data are presented in Table S4. A bare stainless-steel foil substrate exhibited a negligible absorption capacity. The prefixes E- and N-represent $^{10}B$ enriched $B_4C$ and natural $B_4C$, respectively.

films was measured by $I/I_0$, where $I_0$ and $I$ are the intensities of the incident neutron beam and the transmitted beam through the thickness direction of the samples, respectively. The results showed that the neutron absorption capacity, which is the attenuation percentage verified by the permeability, increased almost linearly, even with a small amount of $B_4C$ in the film (Fig. 4a). For electrically neutral neutrons, the main absorption mechanism of the $^{10}B$ isotope in $B_4C$ can be facilitated by (n, α) reactions converting them into charged alpha particles accompanied by a gamma (γ) emission:

$$^{10}B + n_{th}(25\,\text{meV}) \rightarrow {}^4He^{2+}(\alpha) + {}^7Li^{3+} + 2.79\,\text{MeV}(6\%) \quad (1)$$

$$^{10}B + n_{th}(25\,\text{meV}) \rightarrow {}^4He^{2+}(\alpha) + {}^7Li^{3+}\,2.31\,\text{MeV} + 0.48\,\text{Me}(\gamma)(94\%) \quad (2)$$

where $n_{th}$ is a thermal neutron with a kinetic energy of about 25 meV[21]. The pure $Ti_3C_2T_x$ film without $B_4C$ and PVA showed limited ability to shield from neutron radiation, given that the $I/I_0$ value was as high as ≈0.999, which may be due to the relatively low thermal neutron cross-section of titanium and the low hydrogen content compared with that of PVA[43,44]. This suggests that the addition of $B_4C$ significantly improves the neutron radiation shielding properties of the hybrid films as expected with the high absorption reaction of $^{10}B$. In addition, the macroscopic absorption cross-section is the most important parameter for neutron radiation shielding, which determines the interaction probability of target materials with neutrons. To quantitatively evaluate the shielding effectiveness divided by the material thickness, the macroscopic cross-sections for the MBP hybrid film with varying $B_4C$ content were calculated using the Beer–Lambert law given by[45]

$$I = I_0 e^{(-\Sigma_t \cdot d)} \quad (3)$$

where $\Sigma_t$ is the macroscopic cross-section and $d$ is the thickness of the neutron-shielding coating. Notably, the macroscopic cross-section values of our filtrated hybrid films gradually increased with an increasing fraction of $B_4C$, indicating that the particles with high neutron absorbing cross-sections were uniformly distributed in $Ti_3C_2T_x$/PVA films without significant structural deformations such as the change in thicknesses (i.e., 20 wt%: ≈33.67 μm; 40 wt%: ≈41.04 μm; 60 wt%: ≈39.79 μm). This behavior is in good agreement with the observations of cross-sectional morphologies and the results of mechanical testing. In addition, Monte Carlo N-Particle (MCNP) simulations were conducted to examine the transmission probabilities of MBP composite films using a realistic scattering and physics model and to compare the obtained results with experimental measurements (see details in the "Methods" section and Supplementary Fig. 22). The simulated values of absorption capacity for various $B_4C$ contents exhibited a high degree of consistency with the experimental results, thereby demonstrating the accuracy and reliability of the MCNP simulation in assessing the shielding effectiveness of the MBP composite films.

Upon comparing the variation of absorption capacity with the thickness of our hybrid films with those of previously reported materials[17,21,23,24,34,46–51], our films were found to have excellent neutron-shielding performances at ultralow thicknesses (Fig. 4c). Moreover,

with the aligned structure of the deposited MXene flakes being similar to the freestanding hybrid films, our painted hybrid film with a large area of approximately 40 mm × 40 mm on the stainless-steel foil exhibited an excellent macroscopic cross-section comparable to that of a freestanding film having the same fraction of $B_4C$. Based on the similar macroscopic cross-section ($\approx 85.02 \pm 9.32$ cm$^{-1}$) throughout the hybrid films with increasing thickness, the half-value layer (HVL), which represents the shielding thickness at half of the incident neutron count rate, was demonstrated via a repetitive coating process on a bare stainless-steel foil. The stainless-steel substrates did not affect the neutron shielding effectiveness due to their negligible absorption capacity. Notably, the fitted HVL of our filtrated and painted hybrid films is in good agreement with that obtained by the following equation[49]:

$$HVL = \frac{Ln(2)}{\Sigma_t} \qquad (4)$$

which suggests a structural uniformity over a wide thickness range.

Moreover, to ensure a more precise and unbiased comparison with the findings of the other studies based on diverse isotopes of B, we extended our simulations to include scenarios in which different types of $B_4C$ powder were employed. Specifically, we denoted the simulation results as E-MBP, which represents MBP 40 with $^{10}B$ isotope-enriched $B_4C$, and N-MBP, which represents MBP 40 with natural $B_4C$. These composites were characterized by distinct weight fractions of the $^{10}B$ isotope compared with those of the $^{11}B$ isotope, as shown in the calculations provided in Supplementary Table 4. N-MBP consistently exhibited a decrease in the macroscopic cross-section at all $B_4C$ contents (i.e., 20, 40, and 60 wt%), compared with the experimental results. This reduction is attributable to the inherently low neutron absorption of $^{11}B$, which is the predominant isotope in natural $B_4C$, as confirmed in a previous theoretical study[52]. The fitted neutron absorbance with the increasing thickness of N-MBP demonstrated comparable or slightly higher capacities compared with those of conventional composites, thereby suggesting the utilization of MBP with natural $B_4C$. In addition, E-MBP, which was simulated using the same $B_4C$ ratio as the actual MBP, exhibited a trend similar to that of the experimental results. This suggests structural uniformity and good agreement between the MCNP simulation and experimental findings.

As a more realistic parameter for evaluation of the neutron radiation shielding performance, the considerably lower HVL of our filtrated ($\approx 40$ μm) and painted ($\approx 140$ μm) MBP hybrid films than those for other boron-containing structural materials of dissimilar categories showed the possibility for light-weight and multi-functional MXene coating applications (Supplementary Table 5). The freestanding MBP hybrid films, with a density of ~2.0 g cm$^{-3}$ within the $B_4C$ weight fraction range of 20–60 wt% (as shown in the calculations in Supplementary Table 6), can settle on the tips of a dandelion (Fig. 1d). Results of the thermogravimetric analysis (TGA) demonstrated that the hybrid films exhibited thermal stability below ~180 °C, with minimal weight loss below 5%, except for the thermal decomposition of the PVA binder (Supplementary Fig. 23).

## Discussion

We experimentally demonstrate that a hybrid MXene film with homogeneously distributed $B_4C$ particles exhibits mechanical flexibility and anomalous neutron-shielding properties because solution-processable, layered 2D MXene flakes are an ideal robust and high-capacity matrix for $B_4C$ particles. The $B_4C$ filler was incorporated into the continuous network of layered 2D MXene flakes with enhanced interfacial bonding, which enabled the use of a wide range of $B_4C$ contents (i.e., 0–60 wt%) and confirmed the structural stability of the nanomaterial-based shielding composite films. In addition, a facile and scalable neutron-shielding MXene/$B_4C$ hybrid paint is demonstrated.

This shielding composite is particularly valuable because it can be directly integrated with various substrates, rendering it promising for use in wearable and lightweight applications.

In this study, our focus was on resolving the inherent challenges of integrating a high concentration of $B_4C$ within an ultrathin layer. Maintaining structural stability and uniformity under such conditions poses considerable difficulties. Commercially available neutron-shielding materials are typically millimeter-scale thick and occasionally exhibit low B loading, as indicated by product specifications. For instance, commercial products such as Mirrobor™ and SWX-238 have thicknesses of 2 mm and SWX-238 contains ~27.6 wt% B for a thickness of 3.2 mm. We precisely controlled the nanostructures, particularly the size of MXene flake and $B_4C$ particles, and the interaction between the matrix and filler materials to ensure structural integrity. Additionally, the film was obtained using straightforward vacuum filtration or painting processes, eliminating the need for added pressure. This approach enabled us to fabricate films that span several hundred square centimeters, demonstrating a considerable coverage area. The simplicity and scalability of these fabrication techniques hold practical advantages, facilitating the production of films with desired thickness and coverage and spanning large areas without requiring intricate equipment or processes. This work potentially expands the use of diverse MXene coatings with proven performance of EMI shielding[29] and demonstrates their promise for practical applications in various fields.

## Methods

### $Ti_3AlC_2$ synthesis

TiC (Alfa Aesar, 99.5%, 2 μm), Ti (Alfa Aesar, 99.5%, 325 mesh), and Al (US Research nanomaterials, 99.7%, 30 μm) powders were ball-milled with a molar ratio of 2:1:1 for 24 h. The mixed powder was then placed into an alumina crucible, heated at a rate of 5 °C min$^{-1}$ to 1450 °C, and held for 2 h under an Ar flow. The sintered block was ground and sieved through a 325 mesh for further use.

### $Ti_3C_2T_x$ synthesis

2 g of $Ti_3AlC_2$ was slowly poured into a mixed etchant containing 4 mL of HF (Sigma Aldrich, 48%), 12 mL of $H_2O$, and 24 mL of HCl (Alfa Aesar, 36.5–38.0%) and stirred for 24 h at 35 °C to produce a multilayer MXene. The etched mixture was then washed with DI water via centrifugation repeatedly at 3500 rpm for 5 min until the supernatant pH reached ~6 using a 250-mL centrifuge tube. 2 g of LiCl was then dissolved in 40 mL of DI water and vigorously stirred with the multilayer MXene for 12 h. After 12 h, the Li-intercalated multilayer MXene was washed through centrifugation at 3500 rpm for 5 min until self-delamination occurred. When the self-delamination occurred, the dark supernatant was collected repeatedly until the supernatant was diluted. The collected MXene solution was then separated through centrifugate at 3500 rpm for 30 min and the supernatant was concentrated at 12,000 rpm for 10 min for further use.

### n-$B_4C$ preparation

2 g of AR-$B_4C$ was dispersed in DI water and sonicated under an ice bath to prevent harsh oxidation for 2 h. The dispersion was then centrifuged at 4000 rpm for 15 min to settle down the large $B_4C$ particles. The supernatant, which is mainly composed of a small particle (<300 nm), was collected and dried under vacuum at 60 °C for 12 h for further use.

### MBP hybrid freestanding film fabrication

With a fixed weight of 50 mg, MXene and n-$B_4C$ were mixed in different weight ratios. 10, 7.5, and 5 mL of 4 mg mL$^{-1}$ MXene were mixed with 10, 20, and 30 mL of 1 mg mL$^{-1}$ n-$B_4C$ to prepare MB 20, 40, and 60, respectively. As both MXene and n-$B_4C$ have similar negative zeta potentials, the mixed hybrid solution was achieved without further stirring process. A 20 mg mL$^{-1}$ PVA (Mw 85,000–124,000, 99+%

hydrolyzed, Sigma Aldrich) aqueous solution was then mixed into the MB hybrid solution to provide 20 wt% of PVA between MXene and PVA and stirred for 1 h. The mixed MBP hybrid solution was then vacuum-filtrated on a polycarbonate membrane filter (GVS, pore size: 0.1 μm). After the solvent was removed, the MBP hybrid film was peeled off from the membrane filter, and the freestanding MBP film was dried under the ambient atmosphere.

## MBP hybrid painted film fabrication

The process is identical except for the concentration of each solution. 5 mL of 20 mg mL$^{-1}$ MXene was mixed with 5 mL of 20 mg mL$^{-1}$ n-B$_4$C, and then 1 mL of a 100 mg mL$^{-1}$ PVA solution was mixed and vigorously shaken for 15 min. The hybrid paint was then sonicated for 5 min to homogenization and placed into a vacuum for 1 h to move out bubbles from PVA for blade-coating. Stainless-steel and glass substrates were cleaned with acetone, IPA, and DI water and UVO-treated for 15 min. The hybrid paint was blade-coated on the stainless-steel, glass, and nylon membrane, and then dried under the ambient atmosphere.

## Characterization

The XRD patterns were recorded using a high-power XRD instrument (Rigaku, D/MAX2500V/PC) with Cu K$_\alpha$ radiation (40 kV, 200 mA) with a scan step of 0.02°. The morphologies and cross-sections of the samples were studied by cold-field-emission SEM (Hitachi, S-4800). Transmission electron microscopy (TEM) images and selected area electron diffraction (SAED) patterns were acquired using a FETEM instrument (FEI, Tecnai G2 F20 X-Twin). Atomic force microscopy (AFM) images were acquired using a Bruker Dimension AFM instrument operating in the tapping mode. The X-ray photoelectron spectroscopy data were collected using a ThermoFisher K-alpha instrument. Fourier-transform infrared spectroscopy (FTIR) spectra were acquired by a Varian 670 instrument. A dynamic light scattering (DLS) (Zeta sizer, Malvern, Nano ZS) analysis was carried out to obtain the size distribution and zeta potential. The mechanical properties were measured using freestanding films cut into strips (5 mm × 20 mm) for the tensile test. More than five strips per sample were tested using an LS1 material testing machine (Lloyd) with a 10 N load cell. The strips were pulled at a rate of 0.01 mm s$^{-1}$. The specimens were held at a grip distance of ≈10 mm. The tensile strength was calculated with the maximum stress and cross-sectional areas of the strips before failure. For the bending test, the hybrid film/nylon membrane sample cut into a strip with sizes of 15 mm × 40 mm was mounted onto grips, which were repeatedly bent in one direction while establishing the electrical connection. The surface area and pore volume of the samples were obtained using the N$_2$ adsorption/desorption isotherms at 77 K on a physisorption analyzer (ASAP 2420, Micromeritics Instruments). The samples were degassed at 80 °C for 12 h before analysis to remove moisture or any adsorbed contamination. The weight percentage of each element (Ti, B, C, H, and O) in the MBP composite was calculated by assuming the presence of only one Ti$_3$C$_2$ block. The mass densities of the MBP composites were calculated using the following procedure for the MCNP simulations: All MBP samples were cut into 20 mm × 20 mm rectangles, and the actual weight of each sample was measured. To calculate the volume, the thicknesses of the samples were characterized using cross-sectional SEM images. Finally, the mass densities of the MBP composites were determined by dividing the weight by the volume.

## Neutron-shielding ability test

The test was performed by the Korea Research Institute of Standards and Science (KRISS) laboratory using an SP9 $^3$He proportional counter (Centronic, Ltd.) and $^{241}$Am–$^9$Be source with a neutron emission rate of 1.227 × 10$^7$ s$^{-1}$. The counter was set up in the standard thermal neutron field constructed using graphite piles. The detector was shielded with a Cd filter except for the front side, and then the counting rate for total

neutrons ($R_t$) was measured. Subsequently, the counting rate for epithermal neutrons (>0.6 eV) was measured while covering the front of the detector with the Cd filter ($R_e$). The counting rate for thermal neutrons that penetrated the samples was measured by iterating the identical processes. The neutron permeability could be calculated as

$$P = \frac{S_t - S_e}{R_t - R_e},$$

where $S_t$ and $S_e$ are counting rates for total neutrons and epithermal neutrons that penetrated the samples, respectively.

## Monte Carlo N-Particle (MCNP) simulation

The calculations were conducted to simulate the neutron-shielding ability via numerical simulations (MCNP version 6.1)[53]. A circular planar neutron source and a square planar neutron counter were positioned parallel to the MBP hybrid film. The diameter of the neutron source plane and the length of the square neutron counter were both 2 cm. The Maxwell–Boltzmann energy distribution was assumed for the neutron source at 311 K because thermal neutrons emitted from the $^{241}$Am–$^9$Be source typically follow this distribution. Neutrons that escaped beyond the defined boundaries were excluded from the calculations. A two-dimensional diagram of the simulation geometry is shown in Supplementary Fig. 22. The elemental composition, bulk density, and thickness of the MBP hybrid films (MBP20, MBP40, and MBP60) were used as variable parameters for the calculations and are listed in Tables S4 and S6. The histories of $1 \times 10^8$ neutrons were simulated. The simulation tracked the number of neutrons penetrating the shielding material and entering the detection field. The transmission probability of the MBP hybrid films was deduced from the simulation results.

## Data availability

Relevant data supporting the key findings of this study are available within the article and the Supplementary information file. All raw data generated during the current study are available from the corresponding authors upon request.

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

## Acknowledgements

This work was supported by National Research Foundation (NRF) of Korea (Grant Nos. 2019M2D2A1A02059152 and 2021R1C1C1012470) funded by the Ministry of Science, ICT, and Future Planning, by the Ulsan National Institute of Science and Technology (UNIST) (1.190093.01 and 1.220125.01), and by the Korea Research Institute of Chemical Technology (KRICT) (SS2241–10). Some of the data from this manuscript was previously published in the thesis of the author (S.-H.S.)[54].

## Author contributions

J.-H.H. and S.-H.S. prepared materials and performed most of the experiments with assistance from Y.H.J, J.P., H.U.Y., Y.S., Y.C., J.W.; J.-H.B., G.-Y.O.; W.L., and S.H.P. conducted the mechanical measurements; Y.L. and J.H.K. conducted MCNP simulation; I.-C.B. and J.H.K contributed many useful suggestions related to the neutron-shielding measurements and applications; J.-H.H., S.-H.S. and S.-Y.K. wrote the manuscript with the input of all other authors; all authors discussed the results and commented on the manuscript; S.-Y.K. supervised the project.

## Competing interests

The authors declare no competing interests.
