## [Peer Review File · Nature Communications]

Reviewers' Comments:

Reviewer #1:

Remarks to the Author:

Key Results:

The authors made a report on synthesis of Ti₃AlC₂ MAX phase and chemically exfoliated Ti₃C₂T_x MXene, and subsequently prepared hybrid solutions with the MXene flakes, size-selected isotope-enriched ¹⁰B₄C powder, and binder/solvent PVA in different concentrations. The MXene/B₄C/PVA solutions were thereafter vacuum-filtrated or blade-coated into MBP composite films, as referred to by the authors. The authors showed and claimed the MBP films were mechanically flexible and moderately robust, and the B₄C powders were homogeneously distributed within the MXene matrix. The most important claim was that the neutron shielding results showed MBP films' macroscopic (neutron capture) cross-section is one of the best values reported for synthetic B₄C composites to date.

Validity:

The authors provided detail synthesis and characterization information of the Ti₃AlC₂ MAX phase, the Ti₃C₂T_x MXene, and the MXene/PVA solution so one can be convinced with their results related to MAX/MXene. However, all of above steps have been reported before (as the references given by the authors), which justifies the decision of putting the synthesis and characterization in the supplementary information. Hence, the focus of the review is put mainly on the quality of the MBP solutions/films along with their performance in neutron radiation shielding.

The major problem is the comparison of the macroscopic cross-section between the reported values in this work and values from various references, e.g. shown in Figure 4 and Table S4. Nearly all referenced values (Ref. 15, 19, 21, 22, 32, 45-49) were done without specifying the use of "isotope enriched 10-B or 10-B-compounds" in their experimental or theoretical works, and one would assume "non-isotope enriched B or B-compounds" to be the feasible/economical option for the groups in the references. In Ref. 44, two products are commercial (Boral™ and Metamic™) and hence difficult to assess if isotope-enriched raw materials are used or not. Since the natural abundance of 10-B isotope is about 20 at.%, while the author of this work is using 10-B powder with >95% isotope enrichment (Supplementary Note 2), the difference in neutron capture cross-section would be about 5 times without going into detail calculation and considering the scattering of other elements. Such difference is backed by a theoretical work done by Backis et al., EPJ Techniques and Instrumentation (2022) 9:8, that the difference of the macroscopic cross-section between 99% isotope enriched 10-B₄C and non-isotope enriched B₄C (referred to as "natB₄C" in the article) is about one order (see Figure 3 in the article).

Since the macroscopic interaction cross-section is a multiplication between the microscopic interaction cross-section and the atomic density, the 10-B isotope enrichment in the raw material will have a direct, and nearly linear, impact on the final property. Hence, if one would take the reported values in this work, and multiply them by 20% to match the 10-B concentration with other works, and scale them to a similar weight percentage in the matrices (say 60 wt.%), they become usual values that one can expect from regular 10-B₄C powder. Similarly, the half-value layer thickness is a function of the macroscopic cross-section and therefore is strongly influenced by the enrichment. That also means the data comparison currently presented in the work are misleading to the reader at the first glance, and therefore the article should not be published in current status.

Significance:

As assessed in previous section, the reported neutron capture cross-section is likely not a breakthrough and hence lands not much impact on the performance of neutron shielding.

The report of a new B₄C-containing composite for neutron shielding purpose can be interesting for certain applications, especially with a low H-content matrix instead of conventional polymer-based ones to reduce the high neutron scattering cross-section. However, there is no comparing studies in the report showing the new MBP composite is outperforming any commercially available neutron shielding materials comprising B or B₄C, either flexible (e.g. MirroBor or SWX-238 Flexi-Boron) or not (e.g. Boral or Metamic). It is then difficult to argue the technical advance of the new reported

composite with respect to available ones, especially in the areas that the authors repeatedly claimed – “robust” and “flexible”. The authors need comparison results to show that MBP composites are, for example, mechanically more resilient, or thermally more stable, or less prominent to neutron scattering...etc.

Data and Methodology:

The data are mostly well collected, presented, and referenced if applicable, except the aforementioned issues in Figure 4 and Table S4, where the values of samples with isotope enrichment or not are compared. The data shown is convincing that the authors have produced the reported materials (Ti₃AlC₂ MAX phase, Ti₃C₂T_x MXene, MXene/PVA solutions, MBP solutions, and MBP coatings/films). It has even shown that the authors have studied and attempted to optimize the solutions systems for the best mechanical properties.

An issue to be fixed is the lack of information for the thickness measurements in Figure 3c. It is either not clearly, or not even, written how the thickness were measured from the painted layer of the area. In terms of large area shielding, it is also more interesting to see the uniformity (homogeneity) of the coating in terms of thickness and composition over the coated area, rather than an averaged thickness.

Analytical Approach:

There is no obvious issue of statistics that would influence the publication in current version of manuscript.

Suggested improvements:

1. Fix the data presentation in Figure 4 and Table S4, so there is no misleading information. Make a fair comparison study with literature values of the macroscopic neutron capture cross-section - the authors should look for values done with isotope enriched materials. Otherwise, the authors should present new results done with non-isotope enriched B₄C for the comparison, and/or perform a detail theoretical study of their material (Ti₃C₂T_x/B₄C/PVA). A scientific explanation should be provided if the composite has a higher cross-section than another B₄C-based material with a similar weight percentage, instead of just reporting the values.
2. Provide more results for the other properties of the new composite material, e.g. relevant mechanical properties or thermal stabilities, in comparison with existing materials. Especially the ones with very similar characteristics (flexible B₄C-containing neutron shielding). As for now, it is not clear what the new composite material can provide to the field of neutron shielding since there are already materials reasonably flexible and robust.
3. Provide mass density or atomic density of the MBP composites or films so it is easier to perform a calculation of the theoretical neutron interaction cross-section – not just for capture, but also for the overall interaction like fission and scattering. All interactions are important for the shielding, though the latter are often neglected due to lower values. Provide physical properties of the material will also help successors to assure the quality of their solutions, if one would like to reproduce.
4. More detail regarding the thickness measurements in Figure 3c should be given in a clear way, including how it was done (SEM? AFM?) and how the area of interest was selected. It is the best, but not necessary, to provide a position-thickness relation to show the quality of thickness homogeneity.

Clarity and context:

The texts are in general well-written and the detail of the experiment is provided. The article is easy to follow and structured in a logical fashion.

References:

The authors are suggested to either replace the references in Figure 4 and Table S4 with other works done with isotope enriched ¹⁰B, or redo the sample to match the level of enrichment with current references. Including some theoretical works showing the limits of the macroscopic cross-section of B₄C and B₄C composites will also help the readers to assess the performance of the materials.

Reviewer #2:

Remarks to the Author:

Manuscript ID: NCOMMS-23-05458-T

Title: Neutron radiation shielding with a robust 2D layered MXene matrix for boron carbide fillers

Recommendation: Major revision

Comments: In this manuscript, a two-dimensional (2D) $\text{Ti}_3\text{C}_2\text{Tx}$ MXene hybrid film with homogeneously distributed B_4C particles was prepared by vacuum filtration and blade coating methods. With high loading of B_4C , the hybrid film shows excellent neutron-shielding performance at low thicknesses. The experiment result is interesting and is of some practical interest. In order to further improve the readability and clarity of this work, this reviewer has some comments for authors as listed below.

1. Why B_4C was chosen as the neutron shielding filler. Compared with boron and boron nitride, what are the advantages of B_4C ?
2. Please further explain the interaction between MXene and B_4C . Why MXene was selected as the matrix to help dispersing B_4C . Does other two-dimensional materials have the same function?
3. Please add the SEM photos of the surface and cross section of the film prepared by blade coating, in order to illustrate whether there is any difference between the micro-morphology of the film prepared by blade coating and vacuum filtration.
4. Please add the test result of the mechanical properties of the film prepared by blade coating, in order to determine whether there is any difference between the mechanical properties of the film prepared by blade coating and vacuum filtration.
5. To demonstrate that the MBP hybrid films combine firmly to the substrate, the stripping test is needed.
6. Please add serial numbers to the formulas in the manuscript.
8. The English of this article should be further polished. The authors should review the manuscript carefully to improve readability.

Response to Reviewers' comments: We would like to thank the Reviewers for their insightful comments and valuable suggestions on our manuscript entitled “*Neutron radiation shielding with a robust 2D layered MXene matrix for boron carbide fillers*”. We have made careful revision of our manuscript based on comments suggested by the Reviewers. For ease of tracking, we are assigning numbers the Reviewers' comments as shown below. Comments are reproduced verbatim in italics, followed by our responses.

Referee #1

Key Results: *The authors made a report on synthesis of Ti_3AlC_2 MAX phase and chemically exfoliated $Ti_3C_2T_x$ MXene, and subsequently prepared hybrid solutions with the MXene flakes, size-selected isotope-enriched $10B_4C$ powder, and binder/solvent PVA in different concentrations. The MXene/ B_4C /PVA solutions were thereafter vacuum-filtrated or blade-coated into MBP composite films, as referred to by the authors. The authors showed and claimed the MBP films were mechanically flexible and moderately robust, and the B_4C powders were homogeneously distributed within the MXene matrix. The most important claim was that the neutron shielding results showed MBP films' macroscopic (neutron capture) cross-section is one of the best values reported for synthetic B_4C composites to date.*

Validity: *The authors provided detail synthesis and characterization information of the Ti_3AlC_2 MAX phase, the $Ti_3C_2T_x$ MXene, and the MXene/PVA solution so one can be convinced with their results related to MAX/MXene. However, all of above steps have been reported before (as the references given by the authors), which justifies the decision of putting the synthesis and characterization in the supplementary information. Hence, the focus of the review is put mainly on the quality of the MBP solutions/films along with their performance in neutron radiation shielding.*

The major problem is the comparison of the macroscopic cross-section between the reported values in this work and values from various references, e.g., shown in Figure 4 and Table S4. Nearly all referenced values (Ref. 15, 19, 21, 22, 32, 45–49) were done without specifying the use of “isotope enriched 10-B or 10-B-compounds” in their experimental or theoretical works, and one would assume “non-isotope enriched B or B-compounds” to be the feasible/economical option for the groups in the references. In Ref. 44, two products are commercial (BoralTM and MetamicTM) and hence difficult to assess if isotope-enriched raw

materials are used or not. Since the natural abundance of ^{10}B isotope is about 20 at.%, while the author of this work is using ^{10}B powder with >95% isotope enrichment (Supplementary Note 2), the difference in neutron capture cross-section would be about 5 times without going into detail calculation and considering the scattering of other elements. Such difference is backed by a theoretical work done by Backis et al., *EPJ Techniques and Instrumentation* (2022) 9:8, that the difference of the macroscopic cross-section between 99% isotope enriched $^{10}\text{B}_4\text{C}$ and non-isotope enriched B_4C (referred to as “nat B_4C ” in the article) is about one order (see Figure 3 in the article).

Since the macroscopic interaction cross-section is a multiplication between the macroscopic interaction cross-section and the atomic density, the ^{10}B isotope enrichment in the raw material will have a direct, and nearly linear, impact on the final property. Hence, if one would take the reported values in this work, and multiply them by 20% to match the ^{10}B concentration with other works, and scale them to a similar weight percentage in the matrices (say 60 wt.%), they become usual values that one can expect from regular $^{10}\text{B}_4\text{C}$ powder. Similarly, the half-value layer thickness is a function of the macroscopic cross-section and therefore is strongly influenced by the enrichment. That also means the data comparison currently presented in the work are misleading to the reader at the first glance, and therefore the article should not be published in current status.

Significance: As assessed in previous section, the reported neutron capture cross-section is likely not a breakthrough and hence lands not much impact on the performance of neutron shielding.

The report of a new B_4C -containing composite for neutron shielding purpose can be interesting for certain applications, especially with a low H-content matrix instead of conventional polymer-based ones to reduce the high neutron scattering cross-section. However, there is no comparing studies in the report showing the new MBP composite is outperforming any commercially available neutron shielding materials comprising B or B_4C , either flexible (e.g., MirroBor or SWX-238 Flexi-Boron) or not (e.g., Boral or Metamic). It is then difficult to argue the technical advance of the new reported composite with respect to available ones, especially in the areas that the authors repeatedly claimed-“robust” and “flexible”. The authors need comparison results to show that MBP composites are, for example, mechanically more resilient, or thermally more stable, or less prominent to neutron scattering...etc.

Reply on Validity and Significance: We acknowledge that the reported neutron capture cross-section values, facilitated by isotope-enriched ^{10}B , may not be considered a breakthrough. However, our focus was on resolving the inherent challenges of integrating a high concentration of B_4C within an ultrathin layer. Maintaining structural stability and uniformity under such conditions poses considerable difficulties. Commercially available neutron shielding materials typically possess millimeter-scale thickness and low B_4C loading, as indicated by product specifications. For instance, *MirroborTM* has a mass content of over 83% at 2 or 5 mm, while *SWX-238 Flex Boron* contains more than 27.6 wt.% at 3.2 mm. We precisely controlled the nanostructures, particularly the size of MXene flake and B_4C particles, and the interaction between the matrix and filler materials to ensure structural integrity. Additionally, the film was obtained using straightforward vacuum filtration or painting processes, eliminating the need for added pressure. This approach enabled us to fabricate films that span several hundred square centimeters, demonstrating a considerable coverage area. The simplicity and scalability of these fabrication techniques hold practical advantages, facilitating the production of films with desired thickness and coverage and spanning large areas without requiring intricate equipment or processes.

Data and Methodology: The data are mostly well collected, presented, and referenced if applicable, except the aforementioned issues in Figure 4 and Table S4, where the values of samples with isotope enrichment or not are compared. The data shown is convincing that the authors have produced the reported materials (Ti_3AlC_2 MAX phase, $\text{Ti}_3\text{C}_2\text{T}_x$ MXene, MXene/PVA solutions, MBP solutions, and MBP coatings/films). It has even shown that the authors have studied and attempted to optimize the solutions systems for the best mechanical properties.

An issue to be fixed is the lack of information for the thickness measurements in Figure 3c. It is either not clearly, or not even, written how the thickness were measured from the painted layer of the area. In terms of large area shielding, it is also more interesting to see the uniformity (homogeneity) of the coating in terms of thickness and composition over the coated area, rather than an averaged thickness.

Reply on Data and Methodology: Given the limited number of experimental samples, we recognize the limitations in comparing values with and without isotope enrichment. During our

initial investigation into B₄C hybridization with our synthesized 2D MXene, we encountered difficulties creating structurally stable composites due to morphological and surface characteristic variations. We fabricated the composite using a singular B₄C material procured from an international supplier, following an optimization process to achieve optimal mechanical properties. We acknowledge the significance of conducting a comprehensive comparison with multiple B₄C materials. To address this, we have included validated simulations that supplement the experimental data. The revised manuscript describes the methodology and simulation procedures to ensure transparency and foster reproducibility. By incorporating simulations, we aim to thoroughly understand the factors influencing the observed values in both cases and bolster the precision of our findings.

In the revised manuscript, we have provided detailed information regarding the method used to measure the thickness of the painted layer, ensuring clarity and accuracy in our methodology. Furthermore, we have included SEM images of the coated samples at multiple magnifications to underscore the importance of coating uniformity across a large area. These images demonstrate the homogeneity of the coating, visually affirming the uniformity across the coated area.

Overall, these revisions improve the clarity and comprehensiveness of our data and methodology, allowing for a more accurate and thorough evaluation of our results.

***Analytical Approach:** There is no obvious issue of statistics that would influence the publication in current version of manuscript.*

Suggested Improvements:

1. Fix the data presentation in Figure 4 and Table S4, so there is no misleading information. Make a fair comparison study with literature values of the macroscopic neutron capture cross-section – the authors should look for values done with isotope enriched materials. Otherwise, the authors should present new results done with non-isotope enriched B₄C for the comparison, and/or perform a detail theoretical study of their material (Ti₃C₂T_x/B₄C/PVA). A scientific explanation should be provided if the composite has a higher cross-section than another B₄C-based material with a similar weight percentage, instead of just reporting the values.

Reply on Comment (1): To address this comment, we conducted Monte Carlo N-Particle

(MCNP) simulations using non-isotope enriched B₄C. The simulation results allowed us to perform a comprehensive and unbiased comparison with other B₄C-based materials reported in the literature. We obtained data for macroscopic cross-sections with varying isotopic compositions and presented a detailed analysis and comparison in our revised manuscript.

***Revised manuscript**

(page 8 , line 29) In addition, Monte Carlo N-Particle (MCNP) simulations were conducted to examine the transmission probabilities of MBP composite films using a realistic scattering and physics model and to compare the obtained results with experimental measurements. The simulated values of absorption capacity for various B₄C contents exhibited a high degree of consistency with the experimental results, thereby demonstrating the accuracy and reliability of the MCNP simulation in assessing the shielding effectiveness of the MBP composite films.

(page 9 , line 15) Moreover, to ensure a more precise and unbiased comparison with other the findings of the other studies based on diverse isotopes of B, we extended our simulations to include scenarios in which different types of B₄C powder were employed. Specifically, we denoted the simulation results as E-MBP, which represent MBP 40 with ¹⁰B isotope-enriched B₄C, and N-MBP, which represents MBP 40 with natural B₄C. These composites were characterized by distinct weight fractions of the ¹⁰B isotope compared with those of the ¹¹B isotope, as shown in the calculations provided in Supplementary Table S4. N-MBP consistently exhibited a decrease in the macroscopic cross-section at all B₄C contents (i.e., 20, 40, and 60 wt%), compared with the experimental results. This reduction is attributable to the inherently low neutron absorption of ¹¹B, which is the predominant isotope in natural B₄C, as confirmed in a previous theoretical study.⁵² The fitted neutron absorbance with the increasing thickness of N-MBP demonstrated comparable or slightly higher capacities compared with those of conventional composites, thereby suggesting the utilization of MBP with natural B₄C. In addition, E-MBP, which was simulated using the same B₄C ratio as the actual MBP, exhibited a trend similar to that of the experimental results. This suggests structural uniformity and good agreement between the MCNP simulation and experimental findings.

52 Backis, A. *et al.* General considerations for effective thermal neutron shielding in detector applications. *EPJ Tech. Instrum.* 9(1), 8 (2022).

Figure 4. Neutron-shielding performances of MBP hybrid films. (a) Neutron absorption capacity of MBP hybrid films with different B₄C usages. (b) Neutron absorption capacity and calculated macroscopic absorption cross-section of the hybrid films with varying B₄C content. The analysis involved a comparison between the experimental and simulated results. (c) Macroscopic cross-sections and (d) neutron absorption capacities vs. the thicknesses of reported boron-based composites. Dashed lines in (c) and (d) represent the fitted data. Detailed neutron-shielding data are presented in Table S4. The bare stainless-steel foil substrate, coated with the hybrid paint with a B₄C fraction of 40 wt%, exhibited a negligible absorption capacity.

2. Provide more results for the other properties of the new composite material, e.g., relevant mechanical properties or thermal stabilities, in comparison with existing materials. Especially the ones with very similar characteristics (flexible B₄C-containing neutron shielding). As for now, it is not clear what the new composite material can provide to the field of neutron shielding since there are already materials reasonably flexible and robust.

Reply on Comment (2): Thank you for your comments. We have carefully investigated the properties of existing flexible neutron-shielding materials, i.e., Mirrobor™ and SWX-238 Flex

Boron. Additionally, we appreciate your suggestion to provide more results for the other properties of our MBP composite material based on comparison with these existing materials.

We would like to highlight that one of the key advantages of our MBP composite films is their extremely thin thickness compared with materials with millimeter-scale thickness, as indicated by the MSDS files provided for MirroborTM and SWX-238 Flex Boron. This thickness was achievable owing to the 2D nature of the MXene used as the matrix, which enabled a highly aligned structure. For mechanical properties, we have conducted additional tests, specifically tensile test of commercially available neutron-shielding material (Mirrotron, MirroborTM), to evaluate the mechanical properties of our MPB film. The results exhibit higher mechanical strength and resilience compared to the existing materials, promising superior mechanical stability. In addition, thermogravimetric analysis results showed that the MBP film exhibited high thermal stability. i.e., it successfully withstood temperatures of up to 600 °C without structural decomposition. This indicates its potential for achieving excellent performance at elevated temperatures.

In our revised manuscript, we have made the necessary corrections and included additional results to enhance the understanding of the advantages offered by our MBP composite material.

***Revised manuscript**

(page 5, line 22) When compared to a commercially available resin-based shielding material (Mirrotron, MirroborTM), our MBP film showcased lower thickness and higher mechanical strength, suggesting a reduced mechanical degradation rate as a coating material (Supplementary Figure S15).

(page 9, line 35) The freestanding MBP hybrid films, with a density of ~ 2.0 g/cm³ within the B₄C weight fraction range of 20–60 wt.% (as shown in the calculations in Supplementary Table S6), can settle on the tips of a dandelion (Supplementary Figure S22a). Results of the thermogravimetric analysis (TGA) demonstrated that the hybrid films exhibited thermal stability below ~ 180 °C, with minimal weight loss below 5%, except for the thermal decomposition of the PVA binder (Supplementary Figure S22b).

Figure S15. Comparison with a commercially available neutron shielding material. (a) Photograph of MBP 40 and Mirrobor™. (b) Stress-strain curves of Mirrobor™.

Figure S22. Light weightness and thermal stability of MBP hybrid films. (a) Photograph showing light weightness of MBP hybrid film, which can settle on the top of a dandelion. (b) TGA curves of MBP hybrid films with varying B₄C contents and a PVA film.

3. Provide mass density or atomic density of the MBP composites or films so it is easier to perform a calculation of the theoretical neutron interaction cross-section – not just for capture, but also for the overall interaction like fission and scattering. All interactions are important for the shielding, though the latter are often neglected due to lower values. Provide physical properties of the material will also help successors to assure the quality of their solutions, if one would like to reproduce.

Reply on Comment (3): In the revised manuscript, we provided the weight percentages of each element (Ti, B, C, H, and O) in the MBP composite, based on assuming the presence of

only Ti_3C_2 blocks, which is relevant to the MCNP thermal neutron absorption simulation (Supplementary Table S4). In addition, we included the mass density of the MBP composites by providing a detailed procedure for calculating the mass density using the measured weights and cross-sectional SEM measurements of the sample thickness. The calculated mass densities are presented in Supplementary Table S6.

***Revised manuscript**

(page 9, line 19) These composites were characterized by distinct weight fractions of the ^{10}B isotope compared to the ^{11}B isotope as calculated in Supplementary Table S4.

(page 9, line 35) The freestanding MBP hybrid films, with a density of $\sim 2.0 \text{ g/cm}^3$ within the B_4C weight fraction range of 20–60 wt.% (as shown in the calculations in Supplementary Table S6), can settle on the tips of a dandelion (Supplementary Figure S22a).

(page 12, line 24) The weight percentage of each element (Ti, B, C, H, and O) in the MBP composite was calculated by assuming the presence of only one Ti_3C_2 block. The mass densities of the MBP composites were calculated using the following procedure for the MCNP simulations: All MBP samples were cut into $20 \text{ mm} \times 20 \text{ mm}$ rectangles, and the actual weight of each sample was measured. To calculate the volume, the thicknesses of the samples were characterized using cross-sectional SEM images. Finally, the mass densities of the MBP composites were determined by dividing the weight by the volume.

Table S4. Calculated elemental weight fractions of MBP hybrid films.

Samples	¹⁰ B (wt%)	¹¹ B (wt%)	H (wt%)	C (wt%)	O (wt%)	Ti (wt%)
MBP 20	15.18	0.47	1.40	24.11	7.43	51.40
E-MBP MBP 40	30.37	0.94	1.40	25.59	7.43	34.27
MBP 60	45.55	1.40	1.40	27.07	7.43	17.13
MBP 20	3.08	12.58	1.40	24.11	7.43	51.40
N-MBP MBP 40	6.15	25.15	1.40	25.59	7.43	34.27
MBP 60	9.23	37.73	1.40	27.07	7.43	17.13

Table S6. Mass densities of MBP hybrid films.

Category	MBP 20	MBP 40	MBP 60
Thickness (μm)	35.94 ± 2.26	38.81 ± 1.98	36.22 ± 2.71
Volume (cm^3)	0.014376	0.015524	0.014488
Weight (g)	0.0288	0.0302	0.0292
Density (g cm^{-3})	2.00334	1.94537	2.01546

4. More detail regarding the thickness measurements in Figure 3c should be given in a clear way, including how it was done (SEM? AFM?) and how the area of interest was selected. It is the best, but not necessary, to provide a position-thickness relation to show the quality of thickness homogeneity.

Reply on Comment (4): We appreciate your suggestion to provide more detail regarding the

thickness measurements. Thickness measurements were performed using a digital blade micrometer. To accurately measure the thickness of the painted films, we subtracted the thickness of the stainless-steel foil from the total measured thickness. To investigate the thickness across the entire sample area, we randomly selected 20 different points and measured the thickness at each point. We have revised the manuscript to include this information and to clarify the methodology used.

***Revised manuscript**

(page 20, line 5) The thickness was randomly measured at 20 different points on each sample using a digital blade micrometer.

Clarity and context: The texts are in general well-written and the detail of the experiment is provided. The article is easy to follow and structured in a logical fashion.

References: The authors are suggested to either replace the references in Figure 4 and Table S4 with other works done with isotope enriched ^{10}B , or redo the sample to match the level of enrichment with current references. Including some theoretical works showing the limits of the macroscopic cross-section of B_4C and B_4C composites will also help the readers to assess the performance of the materials.

Reply: We have carefully considered your recommendations and thus revised the manuscript considerably. To ensure a more accurate and comprehensive comparison with previous studies, we extended our simulations to include different types of B_4C powders with varying weight fractions of the ^{10}B isotope compared with the ^{11}B isotope. We denote these simulation scenarios as E-MBP and N-MBP, which represent the use of ^{10}B isotope-enriched B_4C with a weight fraction of 32.4 and natural B_4C with a weight fraction of 0.24, respectively. This allowed us to account for isotopic variations and performed unbiased comparisons with references.

We observed a consistent decrease in the macroscopic cross section of N-MBP at different B_4C contents, which was not indicated in the experimental results. This reduction is attributed to the inherently low neutron absorption of ^{11}B , which is the predominant isotope in natural B_4C . Subsequently, we analyzed the fitted neutron absorbance of N-MBP with different thicknesses and discovered comparable or slightly higher capacities compared with those of conventional composites. This indicates the potential of N-MBP with natural B_4C as an

effective neutron-shielding material.

Additionally, the simulated results for E-MBP, based on the same B₄C ratio as the actual MBP, exhibited a trend similar to the experimental results. This indicates structural uniformity and good agreement between the simulation and experimental findings.

Referee #2

Comments: In this manuscript, a two-dimensional (2D) Ti₃C₂T_x MXene hybrid film with homogeneously distributed B₄C particles was prepared by vacuum filtration and blade coating methods. With high loading of B₄C, the hybrid film shows excellent neutron-shielding performance at low thicknesses. The experiment result is interesting and is of some practical interest. In order to further improve the readability and clarity of this work, this reviewer has some comments for authors as listed below.

Reply: We appreciate the reviewer's comment and the request for additional results on the properties of our new composite material. In the revised manuscript, we have included information on the surface and cross-sectional morphologies with element distribution analysis as well as mechanical test results, to further characterize our Ti₃C₂T_x MXene hybrid films. The provided morphological and elemental analysis, along with mechanical property data, enhance the understanding of the structural characteristics and composition of our painted MBP films, further validating their potential as flexible and robust neutron-shielding materials.

1. Why B₄C was chosen as the neutron shielding filler. Compared with boron and boron nitride, what are the advantages of B₄C?

Reply on Comment (1): We would like to appreciate to the referee's comment. Boron naturally exists as two stable isotopes, i.e., ¹⁰B and ¹¹B, at a ratio of 1:4, and its neutron capture ability is mainly attributed to isotope ¹⁰B. Therefore, boron and boron compounds such as boron oxide (B₂O₃) and boron nitride (BN) are promising candidates for neutron radiation shielding. However, the surface density of boron, which is directly related to the neutron-shielding capabilities of boron compounds, is prominent in the order of B₂O₃, h-BN, B₄C, and B (Analytica Chimica Acta, 124, 373, (1981)) and (P. Tamayo, (2020) Micro and Nanostructured Composite Materials for Neutron Shielding Applications, Woodhead Publishing). Although boron has a higher neutron-shielding ability than B₄C, many impurities in commercially

available boron powder produce secondary gamma emissions after irradiation. In this regard, B₄C is the most widely used form in neutron-shielding applications. Moreover, the chemical inertness and low price of B₄C (B₄C: 1.76 \$/g, BN: 3.72\$/g, and B: 3.85\$/g from Sigma Aldrich) render it a conventional neutron-absorbing/shielding material for many nuclear industries, as described in the main text.

The synthesis of pure boron is extremely difficult because of its high reactivity and extreme hardness, which can result in impurities or superficial oxidation. By contrast, boron nitride is not a significant neutron absorber and generally exhibits a smaller macroscopic cross-section than B₄C.

In response to the reviewer's questing regarding the advantages of B₄C, we have made the necessary corrections in the introduction part and inserted new references of 14-15.

***Revised manuscript**

(page 2, line 23) Boron carbide (B₄C) is widely used as a neutron-absorbing material because of its high reaction cross-section of ¹⁰B, high melting point of 2,763 K, and usefulness for neutron capture.^{12,13} By contrast, pure boron and other boron compounds (such as hexagonal boron nitride) exhibit serious purity issues due to its high reactivity and inadequate shielding properties, respectively.^{14,15}

14 Özdemir, T & Yılmaz, S. N. Hexagonal boron nitride and polydimethylsiloxane: A ceramic rubber composite material for neutron shielding. *Radiat. Phys. Chem.* **152**, 93-99 (2018).

15 Bem, H., & Ryan, D.E. Choice of boron shield in epithermal neutron activation determinations. *Analytica Chimica Acta*, **124**, 373-380, (1981)

2. Please further explain the interaction between MXene and B₄C. Why MXene was selected as the matrix to help dispersing B₄C. Does other two-dimensional materials have the same function?

Reply on Comment (2): As described in Supplementary Note 3, the interaction between MXene and B₄C in the colloidal dispersion is primarily governed by electrostatic repulsion. owing to the negative surface charge of both materials (Supplementary Figure S10). However, in the case of large B₄C (L-B₄C) particles, an electrostatic attraction force between MXene and L-B₄C was observed, which resulted in the aggregation of MXene flakes on the surface of L-B₄C (Supplementary Figure S3). This phenomenon is attributed to the lower presence of boron

oxide on the surface of L-B₄C, which eliminates the negative surface potential typically associated with B₄C. We have now included supplementary data, specifically the surface potential of L-B₄C particles, in Supplementary Figure S10a.

Whereas other 2D materials with negative zeta potentials can potentially serve as matrices for B₄C, the large flake size of MXene renders it suitable for accommodating B₄C within its structure. In our case, the synthesized MXene flakes have an average diameter of approximately 5 μm (Supplementary Figure S1), which enhances the mechanical properties of the resulting films. To the best of our knowledge, other 2D materials such as h-BN and MoS₂ (which are synthesized via top-down liquid exfoliation method) typically have sub-micron flake sizes and lower production yields, which limit their utility as matrices for B₄C and their commercialization potential.

***Revised manuscript**

(Supplementary Information, page 5, line 9) To investigate the effect of the particle size on the interaction between Ti₃C₂T_x and B₄C, large B₄C (L-B₄C) particles were isolated via centrifugation and decantation. The ζ potential of L-B₄C exhibited a broader and higher distribution compared with those of AR-B₄C and n-B₄C. Despite the negative average value of L-B₄C, a positive region was clearly observed in the ζ potential, which suggests the presence of an electrostatic attraction force between L-B₄C and MXene. This force results in the aggregation of MXene flakes on the surfaces of larger B₄C particles, as shown in Supplementary Figure S3.

Figure S10. Zeta potential analysis. (a) AR-B₄C (black), n-B₄C (red), and L-B₄C (blue). (b) n-B₄C (black), Ti₃C₂T_x MXene (red), and MB hybrid (blue).

3. Please add the SEM photos of the surface and cross section of the film prepared by blade

coating, in order to illustrate whether there is any difference between the micro-morphology of the film prepared by blade coating and vacuum filtration.

Reply on Comment (3): The surface and cross-sectional morphologies of the painted films, as shown in Supplementary Figure S17, demonstrate that they possess the same structure as the vacuum-filtrated films. This indicates that the blade coating process did not introduce significant changes to the overall morphology and structure of the films.

To confirm the even distribution of the elements within the painted films, we performed EDS mapping analysis on both the surface and cross-section parts. The EDS mapping results, depicted in Supplementary Figure S18, clearly show that Ti, B, and C were evenly distributed throughout the entire area, suggesting that the n-B₄C particles were effectively mixed with the MXene matrix, resulting in a uniform element distribution.

***Revised manuscript**

(page 7, line 4) The surface and cross-sectional morphologies displayed in Supplementary Figure S17 showed that the painted films possessed the same structure as vacuum-filtrated ones. EDS mapping carried out on the surface and the cross-section of painted MBP films further confirmed the even distribution of Ti, B, and C elements throughout the entire area, indicating that the n-B₄C was evenly mixed into the MXene matrix (Supplementary Figure S18).

Figure S17. Morphologies of the painted MBP hybrid film on nylon fabric membrane. SEM images of (a) surface and (b and c) cross-section of the blade-coated MBP hybrid film on a nylon membrane.

Figure S18. EDS mapping images of painted MBP hybrid films on nylon substrate. Images of (a) surface and (b) cross-section of the blade-coated MBP hybrid film with a B₄C weight fraction of 40 wt.% showing the uniform distribution of boron in the film.

4. Please add the test result of the mechanical properties of the film prepared by blade coating, in order to determine whether there is any difference between the mechanical properties of the film prepared by blade coating and vacuum filtration.

Reply on Comment (4): We conducted mechanical property tests on the films prepared by blade coating technique and have included the results in the revised manuscript (Supplementary Figure S20). The mechanical properties of our prepared films were comparable to those of the vacuum-filtered film, indicating that the choice of the coating method did not significantly affect the mechanical strength of the MBP film. Additionally, the mechanical strength and failure strain of the hybrid film were primarily determined by the properties of the substrate, highlighting the importance of using robust and flexible substrates to achieve the desired mechanical performance.

***Revised manuscript**

(page 7, line 11) Furthermore, it can be observed that the mechanical strength of the nylon membrane painted with an MBP film does not exhibit any remarkable changes (Supplementary Figure S20). The mechanical strength and failure strain of the hybrid film are primarily attributed to the material properties of the substrate, enabling it to withstand uniform stress-strain by introducing robust and flexible substrates.

Figure S20. Stress-strain curves of painted MBP hybrid film with a B₄C weight fraction of 40 wt.% on a nylon substrate.

5. To demonstrate that the MBP hybrid films combine firmly to the substrate, the stripping test is needed.

Reply on Comment (5): A stripping test was conducted using adhesive tape to evaluate the bonding strength between the painted MBP films and a nylon substrate. The results of the stripping test confirmed that the film exhibited good adhesion to the substrate, as it remained adhered to the substrate with minimal disruption of the film surface after tape removal. This qualitative assessment provides evidence of the strong bonding between the MBP film and the nylon substrate, further supporting the suitability of our proposed MBP hybrid films for practical applications. We have included the results and corresponding figure (Supplementary Figure S19) in the revised manuscript.

***Revised manuscript**

(page 7, line 8) A stripping test using adhesive tape qualitatively implied good bonding strength between the painted MBP films and the nylon substrate. It showed that the film remained to the substrate with minimal disruption of the film surface after tape removal (Supplementary Figure S19).

Figure S19. Stripping test on the MBP hybrid film painted on a nylon membrane.

6. Please add serial numbers to the formulas in the manuscript.

Reply on Comment (6): In the revised version, we have added the serial numbers to the formulas. We thank the reviewer for pointing it out.

7. The English of this article should be further polished. The authors should review the manuscript carefully to improve readability.

Reply on Comment (7): To further polish the article and improve its readability, we have thoroughly proofed the manuscript for grammar, phrasing, punctuation, and typos.

Reviewers' Comments:

Reviewer #1:

Remarks to the Author:

Remarks to the author:

The authors have answered to the comments of the two reviewers and made changes to the manuscript accordingly. The quality of the paper has increased from the modifications and the additional descriptions and data provided. However, major issues either remained from previous version or appearing from the changes, which have great importance to the key information delivered by the article, should be further discussed. Hence, the manuscript is not suitable for publication in current status:

See comments below:

1. P.2, L23, the added sentences “*Boron carbide (B4C) is widely used as a neutron-absorbing material because of its high reaction cross-section of 10B, high melting point of 2,763 K, and usefulness for neutron capture.*^{12,13} By contrast, pure boron and other boron compounds (such as hexagonal boron nitride) exhibit serious purity issues due to its high reactivity and inadequate shielding properties, respectively.^{14,15}”

First of all, “*usefulness for neutron capture*” is rather an overall comment than a property to describe boron carbide. It also covers the “*high reaction cross-section of 10B*” otherwise it will never become useful, i.e. they are repeated. Please modify this sentence so it lists relevant properties for neutron capture/shielding.

Secondly, I do not agree with the authors that other boron compounds, including h-BN, are not adequate for shielding due to its high reactivity. What reactivity are the authors referring to here, nuclear/neutron/chemical? Please be more careful and clearer when using *reactivity*. To my best knowledge, at least BN is reasonably chemically inert and radiation hard, so it is still a candidate for shielding purpose.

In fact, the new Ref.14 (Özedmir & Yilmaz, 2018) and Ref.15 (Bem & Ryan, 1981) do not agree with the authors' statements regarding other boron compounds for neutron capture purpose. Ref.15 shows that, except pure B, all other compounds have low chemical impurity level comparable with B4C. The main concern with pure B is the metal impurity can be activated after neutron irradiation and subsequently emit gamma-rays (By the way, this is the same problem that the author didn't address about their MBP, where 17 – 51 wt.% Ti can be activated and emit gamma-rays!). Ref.14

already states that “*h-BN has high resistance to chemicals*” in the introduction. The authors need to rework the second sentence to justify the decision of using boron carbide instead of other compounds, in reply to the reviewer’s comment. The information of current text is incorrect.

2. P.7 and Fig.S18, the authors claim “*The EDS mapping results of the surface and cross-section of the painted MBP films further confirmed the even distribution of Ti, B, and C throughout the entire area, indicating that n-B4C was evenly mixed in the MXene matrix (Supplementary Figure S18).*” But in Fig.S18 it seems like the B and C are more concentrated on the surface. If B4C and MXene are evenly mixed as claimed, the mapping of Ti, B, and C should not have such big difference. Please provide more evidence to the statement.
3. Following comment 2, P.20, caption for Fig.3, saying the thickness was acquired randomly from 20 positions does not answer the reviewer’s comment regarding the uniformity of the thickness of this coating. However, the physical properties of the MBP are exactly the important feature of this study, as the authors concluded that they wanted to resolve the challenge of a high-B4C-content thin-layer with desirable properties. Please be more precise with the measurement setup.

Also, from Fig.3d, the thickness of the coatings on all three substrates visually seems to be inhomogeneous, with a varying factor of 2. What is the authors’ interpretation here? Is that a problem from the coating technique, or the solution?

4. P.8, it is hard to accept such brief description for the MCNP calculation performed. The authors need to provide much more information than current status, which simply just said MCNP calculation is done and without any other parameters. What are the codes, the models, and the physical parameters applied?
5. Fig.4c does not present any scientific meaning and therefore is pointless. The reason is that the macroscopic cross-section, Σ , is not a function of thickness:

$$\Sigma(E) = \sigma(E) \cdot N$$

Where σ is the microscopic cross-section (element specific), N is the atomic density (number of atoms per cm^3), and E is the energy of neutrons. The only case where thickness is relevant is when the atomic density is a function of the thickness, i.e. the composition/density of the film is varying with the thickness. If that is the case, the authors need to provide more evidence to support the claim.

Normally, it is the absorption probability (or the absorption capacity in the authors' term), P , that is a function of the material's thickness, d :

$$P(E) = 1 - \exp(-\Sigma(E) \cdot d)$$

Which makes Fig.4d reasonable to plot. In other words, the difference in Fig.4c for different samples is mostly because of the atomic density of ^{10}B in the samples. The suggestion is to remove current Fig.4c.

6. Following comment 5, in the abstract, last sentence, since the macroscopic cross-section of the material is not related to the thickness, there is nothing usual or unusual about the value obtained for the $40\ \mu\text{m}$ film. The authors are suggested to either remove "*unusual for films with small thickness (40 micrometer)*", or replace the macroscopic cross-section with the absorption probability. Please also mention the Am-Be neutron source here for easier comparison.

The following are relatively minor issues:

7. P.10, second paragraph of the conclusion, the authors state that the commercially available materials possess low B₄C loading, but immediately in the next sentence mention that MirroBor™ has a mass content over 83%, which is much higher than the authors 60 wt.% MBP sample and the best performed MBP sample with 40 wt.%. The two sentences possess contradictory information and should be corrected.
8. Table S6, please check the effective number of digits. In the authors' description they cut the samples into "20 mm × 20 mm" pieces, i.e. 0.20 cm by 0.20 cm. This does not automatically come to 5 or 6 effective number of digits for the volume and the density.

Reviewer #2:

Remarks to the Author:

The manuscript has been well revised according to the comment and all the questions have been answered.

The manuscript is suggested to be accepted.

Response to Reviewers' comments: Thank you very much for your valuable feedback on the revised manuscript. We appreciate your insights and would like to address the concerns raised in your review. Here is our revised manuscript, along with a reply to your comments. For ease of tracking, we are assigning numbers the Reviewer's comments as shown below. Comments are reproduced verbatim in italics, followed by our responses.

Reviewer #1

Remarks to the author: The authors have answered to the comments of the two reviewers and made changes to the manuscript accordingly. The quality of the paper has increased from the modifications and the additional descriptions and data provided. However, major issues either remained from previous version or appearing from the changes, which have great importance to the key information delivered by the article, should be further discussed. Hence, the manuscript is not suitable for publication in current status:

Reply: We greatly appreciate the reviewer's insightful comments on our work, as well as his/her suggestions to improve the quality of our paper. We address below the point-by-point reviewer's comments and have made clarifying revisions and expansions to the manuscript.

1. P.2, L23, the added sentences "Boron carbide (B_4C) is widely used as a neutron-absorbing material because of its high reaction cross-section of ^{10}B , high melting point of 2,763 K, and usefulness for neutron capture.^{12,13} By contrast, pure boron and other boron compounds (such as hexagonal boron nitride) exhibit serious purity issues due to its high reactivity and inadequate shielding properties, respectively.^{14,15}"

First of all, "usefulness for neutron capture" is rather an overall comment than a property to describe boron carbide. It also covers the "high reaction cross-section of ^{10}B " otherwise it will never become useful, i.e., they are repeated. Please modify this sentence so it lists relevant properties for neutron capture/shielding.

Secondly, I do not agree with the authors that other boron compounds, including h-BN, are not adequate for shielding due to its high reactivity. What reactivity are the authors referring to here, nuclear/neutron/chemical? Please be more careful and clearer when using reactivity. To my best knowledge, at least BN is reasonably chemically inert and radiation hard, so it is still a candidate for shielding purpose.

In fact, the new Ref.14 (Özedmir & Yilmaz, 2018) and Ref.15 (Bem & Ryan, 1981) do not agree with the authors' statements regarding other boron compounds for neutron capture purpose. Ref.15 shows that, except pure B, all other compounds have low chemical impurity level comparable with B₄C. The main concern with pure B is the metal impurity can be activated after neutron irradiation and subsequently emit gamma-rays (By the way, this is the same problem that the author didn't address about their MBP, where 17–51 wt.% Ti can be activated and emit gamma-rays!). Ref.14 already states that “h-BN has high resistance to chemicals” in the introduction. The authors need to rework the second sentence to justify the decision of using boron carbide instead of other compounds, in reply to the reviewer's comment. The information of current text is incorrect.

Reply on Comment (1): We agree. We thank the reviewer for pointing it out. We have taken your concerns into consideration and made revisions in the manuscript accordingly. Following the reviewer's first comment, we have changed the sentence discussing the "usefulness for neutron capture" by emphasizing the specific properties of B₄C that make it suitable for neutron capture and shielding purposes.

Regarding the second point, we are sorry for any confusion occurred during the last revision process. As the reviewer exactly pointed it out, h-BN possesses desirable properties such as chemical inertness and radiation hardness, which make it a promising material for various shielding applications. In response to the reviewer's questing regarding the advantages of B₄C, we have made the necessary corrections in the introduction part as follows.

***Revised manuscript**

(page 2, line 24) Boron carbide (B₄C) is widely used as a neutron-absorbing material because of its high reaction cross-section of ¹⁰B, high melting point (2,763 K), and low density (2.52 g cm⁻³).^{12,13} Boron naturally exists as two stable isotopes, i.e., ¹⁰B and ¹¹B, at a ratio of 1:4, and its neutron capture ability is mainly attributed to isotope ¹⁰B. Therefore, boron and boron compounds such as boron oxide (B₂O₃) and boron nitride (BN) are promising candidates for neutron radiation shielding. However, the surface density of boron, which is directly related to the neutron-shielding capabilities of boron compounds, is prominent in the order of B₂O₃, hexagonal BN, B₄C, and B.^{14,15} Although B has a higher neutron-shielding ability than B₄C, many impurities in commercially available boron powder produce secondary gamma emissions after irradiation. In this regard, B₄C is the most widely used form in neutron-shielding applications. Therefore,

- 14 Bem, H., & Ryan, D.E. Choice of boron shield in epithermal neutron activation determinations. *Analytica Chimica Acta*, **124**, 373-380 (1981).
- 15 Tamayo, P., Thomas, C., Rico, J., Cimentada, A., Setién, J., & Polanco, J. A. Review on neutron-absorbing fillers. in *Micro and Nanostructured Composite Materials for Neutron Shielding Applications* (ed. Abdulrahman, S. T., Ahmad, Z., & Thomas, S.) 25-52 (Woodhead Publishing, 2020).

With regard to the Ti issue in our MBP samples raised by the reviewer, we first would like to note that the pure $Ti_3C_2T_x$ film (without B_4C and PVA) with small thickness (approximately 40 μm) showed limited ability to shield from neutron radiation, given that the I/I_0 value (i.e., neutron permeability of the 40- μm -thick, pure $Ti_3C_2T_x$ film) was as high as ≈ 0.999 , as described in the main text in lines of 16-18 on page 8. This indicates that the effect of Ti activation and subsequent emission of gamma rays is negligible in our MBP samples with small thicknesses. We note that our focus was on resolving the inherent challenges of integrating a high concentration of B_4C within an ultrathin layer.

For the reviewer's interest, of the Ti isotopes that can be produced by thermal neutrons, only Ti-50 is radioactive. Not only is the probability of its formation (thermal neutron capture cross section) very low [Refs. A & B], but its product, Ti-51, also undergoes a brief period of electron emission (beta decay) with a half-life of 5.76 min, making the activation effect of Ti due to thermal neutrons negligible. Regarding gamma-ray emission of Ti isotopes, the amount of prompt gamma-ray emission that occurs in neutron capture reactions is proportional to their neutron capture cross sections. However, not only is the neutron capture cross section value of Ti very low, that of Ti-49 is also low [Ref. C]. Ti-49 is known to emit high-energy gamma rays, and its probability of neutron capture reactions is approximately 1.6% of the entire Ti isotope [Refs. B & C].

As a result, the nuclear industry considers Ti (e.g., the Ti-6Al-V alloy) as a key element, using it in the structural components of nuclear reactors and spent nuclear fuel storage systems [Refs. D & E]. Recently, Ti has also been investigated as an alloying material of stainless steel, which is a critical structural material in nuclear reactors. This trend is prominent not only in pressurized water reactors but also in nuclear-fusion and advanced reactors (e.g., high-speed reactors), where Ti alloys or Ti-added metal alloys are being considered as major structural materials [Refs. F&G].

- [A] Akerib, D, *et al.* Radio-assay of Titanium samples for the LUX Experiment. <https://arxiv.org/abs/1112.1376v3> (2011).
- [B] Mughabghab, S. F. Thermal neutron capture cross sections resonance integrals and g-factors. International Atomic Energy Agency, International Nuclear Data Committee, Vienna (Austria) (2003).
- [C] Kim, C. K., Meinke, W. W. Thermal neutron-activation analysis of titanium using 5·8-minute titanium-51 and rapid radiochemical separations. *Talanta* **10**, 83-89 (1963).
- [D] Bignon, Q. *et al.* Oxide formation on titanium alloys in primary water of nuclear pressurised water reactor. *Corrosion Science* **150**, 32-41 (2019).
- [E] Mahmud, A. *et al.* Mechanical Behavior Assessment of Ti-6Al-4V ELI Alloy Produced by Laser Powder Bed Fusion. *Metals* **11**, 1671 (2021).
- [F] Jones, R. H., Leonard, Jr. B., Johnson, Jr. A. Assessment of titanium alloys for fusion reactor first-wall and blanket applications. Final report. Battelle Pacific Northwest Labs., Richland, WA (United States) (1980).
- [G] Shin, S. H., Kim, J. H., Kim, J. H. Corrosion behavior and microstructural evolution of ASTM A182 Grade 92 steel in liquid sodium at 650°C. *Corrosion Science* **97**, 172-182 (2015).

We hope the revised manuscript clarifies our statements and provides a more accurate and comprehensive description of B₄C neutron-shielding filler and its advantages for neutron capture and shielding.

2. P.7 and Fig.S18, the authors claim “The EDS mapping results of the surface and cross-section of the painted MBP films further confirmed the even distribution of Ti, B, and C throughout the entire area, indicating that n-B₄C was evenly mixed in the MXene matrix (Supplementary Figure S18).” But in Fig.S18 it seems like the B and C are more concentrated on the surface. If B₄C and MXene are evenly mixed as claimed, the mapping of Ti, B, and C should not have such big difference. Please provide more evidence to the statement.

Reply on Comment (2): We agree and thank the reviewer for careful observation. As the reviewer pointed it out, Supplementary Figure S18 in the previous manuscript indicated certain irregularities in the EDS mapping. We note that these irregularities were attributed to the cutting process employed during the cross-sectional observation.

We performed additional measurements to effectively address this concern and improve the evidence that n-B₄C is evenly distributed in the MXene matrix. The revised manuscript

incorporates Supplementary Figure S18, which presents a collection of EDS mapping images showing MBP samples applied to different substrates. The additional EDS mapping results indicated a notable decrease in structural discrepancies and demonstrated a more homogeneous distribution of Ti, B, and C throughout the MBP coating layers. The observed results showed a higher degree of homogeneity in the mixture of n-B₄C within the MXene matrix. This finding provides additional support for our initial claims.

We cordially request the reviewer to consider the potential impact of the cutting process, as it can accommodate the variability based on factors such as the hardness of the coated substrate, structural characteristics of 2D MXenes, and methodological limitations associated with EDS measurements due to variations in morphology.

We hope that these additional results address the reviewer's concerns and provide a clearer understanding of the distribution of elements in the painted MBP films.

***Revised manuscript**

(page 7, line 10) EDS mapping results of the surface and cross-section of the painted MBP films on various substrates (i.e., stainless-steel, glass, and nylon) further confirmed the even distribution of Ti, B, and C throughout the area, indicating that n-B₄C was evenly mixed in the MXene matrix (Supplementary Figure S18).

Figure S18. Cross-sectional EDS mapping images of painted MBP hybrid films on a various substrate. Cross-sectional images of blade-coated MBP hybrid films on (a) stainless-steel, (b) glass, and (c) nylon substrates with a B₄C weight fraction of 40 wt.% showing the uniform distribution of boron in the film.

3. Following comment 2, P.20, caption for Fig.3, saying the thickness was acquired randomly from 20 positions does not answer the reviewer's comment regarding the uniformity of the thickness of this coating. However, the physical properties of the MBP are exactly the important feature of this study, as the authors concluded that they wanted to resolve the challenge of a high-B₄C-content thin-layer with desirable properties. Please be more precise with the measurement setup.

Also, from Fig.3d, the thickness of the coatings on all three substrates visually seems to be inhomogeneous, with a varying factor of 2. What is the authors' interpretation here? Is that a problem from the coating technique, or the solution?

Reply on Comment (3): We thank the reviewer for providing valuable feedback. In the revised manuscript, we have provided a description of the improved measurement setup to obtain the film thicknesses more accurately. The thickness measurement of the film was performed using a stainless-steel foil, approximately 100- μ m-thick. This foil was used to cover a substantial surface area of 25 mm \times 50 mm. The thicknesses of the painted MBP samples were assessed using a digital micrometer that facilitated measurements at 10 distinct locations. To maintain a high level of precision, specific measurement locations were visually represented in the photograph of the painted MBP sample. This is shown in the insets of Figure 3 and Supplementary Figure A. This method provides a more systematic and uniform approach to measuring the thickness of the coating and addresses the reviewer's concerns regarding thickness uniformity.

The observed variation in the coating film thickness, depending on the substrate, may be attributable to the differences in surface roughness and surface energy. As evidenced by the contact angles shown in Figure 3d in the revised manuscript, the distinct surface energies of each substrate induced varying interfacial energies and van der Waals interactions at the solid-liquid interface. These differences in wettability led to thickness deviations in the manually applied blade-coating method utilized in this study. However, it is anticipated that automated methods, such as doctor-blade coating, which affords better thickness uniformity, or

adjustments in paint concentration, could be optimized to address this issue.

***Revised manuscript**

(page 6, line 35) Moreover, a uniform thickness over a large area of approximately 25 mm × 50 mm was confirmed via measurements at 10 different locations by using a digital micrometer (Figure 3c).

Figure 3. Blade-coating of MBP hybrid paint. (a) Photographs of the painted MBP hybrid film on concave and convex surfaces of a stainless-steel foil. (b) Thicknesses of the painted MBP hybrid films as a function of the number of paintings. (c) Thicknesses of the films painted on stainless-steel foils (thickness $\approx 100 \mu\text{m}$) over a large area of 25 mm × 50 mm. The inset shows an image of the painted MBP sample. The thicknesses were measured at 10 different points on photograph by using a digital blade micrometer. (d) SEM cross-sectional images of the hybrid film on the various substrates including the (i and iv) stainless-steel foil, (ii and v) glass, and (iii and vi) nylon fabric membrane. Contact angles of the bare substrates are included

on the upper right in (i-iii). (e) Photograph of the painted MBP hybrid film on rolled surfaces of the nylon membrane with a large area of $10 \times 30 \text{ cm}^2$. (f) Resistance changes as a function of the bending cycle of the hybrid film of painted 40 wt.% MBP on a nylon membrane at different bending radii. The insets show OM images of the released (left) and bent (right) MBP hybrid film/nylon membrane sample. (g) Magnified view of the yellow dashed area in (f). The average and standard deviation of the thickness in (b) were calculated by 14 measurements within the area of the sample, while those in (c) were calculated by 20 measurements within the area of the sample.

Figure A. Thickness measurement of painted MBP hybrid films on a stainless-steel foil.

4. P.8, it is hard to accept such brief description for the MCNP calculation performed. The authors need to provide much more information than current status, which simply just said MCNP calculation is done and without any other parameters. What are the codes, the models, and the physical parameters applied?

Reply on Comment (4): We agree that we were not clear in our explanation. Detailed descriptions on MCNP codes and models are added in the end of ‘Experimental section’ in the revised manuscript. And the physical parameters which were used in the MCNP calculation are described in Supplementary Tables S4 and S6 in the revised manuscript.

***Revised manuscript**

(page 13, line 14) *Monte Carlo N-Particle (MCNP) simulation:* The neutron-shielding ability was studied via numerical simulations. A circular planar neutron source and square planar neutron counter were positioned parallel to the MBP hybrid film. The diameter of the neutron source plane and length of the square neutron counter were both 2 cm. The Maxwell-Boltzmann

energy distribution was assumed for the neutron source at 311 K because thermal neutrons emitted from the ^{241}Am - ^9Be source typically follow this distribution. Neutrons that escaped beyond the defined boundaries were excluded from the calculations. A two-dimensional diagram of the simulation geometry is shown in Supplementary Figure S22. The elemental composition, bulk density, and thickness of the MBP hybrid films (MBP20, MBP40, and MBP60) were used as variable parameters for the calculations and are listed in Tables S4 and S6. The histories of 1×10^8 neutrons were simulated. The simulation tracked the number of neutrons penetrating the shielding material and entering the detection field. The transmission probability of the MBP hybrid films was deduced from the simulation results.

Figure S22. Schematic of MCNP simulation on MBP hybrid films.

5. Fig.4c does not present any scientific meaning and therefore is pointless. The reason is that the macroscopic cross-section, Σ , is not a function of thickness:

$$\Sigma(E) = \sigma(E) \cdot N$$

Where σ is the microscopic cross-section (element specific), N is the atomic density (number of atoms per cm^3), and E is the energy of neutrons. The only case where thickness is relevant is when the atomic density is a function of the thickness, i.e., the composition/density of the film is varying with the thickness. If that is the case, the authors need to provide more evidence to support the claim.

Normally, it is the absorption probability (or the absorption capacity in the authors' term), P , that is a function of the material's thickness, d :

$$P(E) = 1 - \exp(-\Sigma(E) \cdot d)$$

Which makes Fig.4d reasonable to plot. In other words, the difference in Fig.4c for different samples is mostly because of the atomic density of ^{10}B in the samples. The suggestion is to remove current Fig.4c.

Reply on Comment (5): We appreciate the reviewer's insightful comment and clarification on the scientific meaning of Fig. 4c. We acknowledge that the macroscopic cross-section, Σ , is not a direct function of thickness and that considering the absorption probability, P , as a function of the material's thickness, d , would be more relevant.

Initially, we aimed to demonstrate that the composite maintained structural uniformity and mechanical properties despite varying the B_4C content in the range of 20%–60% by using a macroscopic cross-section. Apparently, the thickness did not significantly change even with this variation in B_4C content, allowing the composite to maintain a significant ^{10}B atomic density.

However, we realized that the current representation might lead to confusion and could be misinterpreted, especially in the absence of a clear correlation between the atomic density and thickness. Moreover, without a direct comparison with other references wherein such investigations might not have been performed, the interpretation could be further complicated.

Taking the reviewer's feedback into account, we have removed the original Fig. 4c in the revised manuscript. We believe this decision will enhance the clarity of the findings and avoid any misinterpretation. We are grateful for the reviewer's valuable input, which has helped us improve the manuscript.

***Revised manuscript**

(page 9, line 7) Upon comparing the variation of absorption capacity with thickness of our hybrid films with those of previously reported materials,^{17,21,23,24,34,46-51} our films were found to have excellent neutron-shielding performances at ultralow thicknesses (Figure 4c).

Figure 4. Neutron-shielding performances of MBP hybrid films. (a) Neutron absorption capacity of MBP hybrid films with different B₄C usages. (b) Neutron absorption capacity and calculated macroscopic absorption cross-section of the hybrid films with varying B₄C content. The analysis involved a comparison between the experimental and simulated results. (c) Neutron absorption capacity vs. thickness of boron-based composites. The dashed lines in (c) represent the fitted data. Detailed neutron-shielding data are presented in Table S4. A bare stainless-steel foil substrate exhibited a negligible absorption capacity.

6. Following comment 5, in the abstract, last sentence, since the macroscopic cross-section of the material is not related to the thickness, there is nothing usual or unusual about the value obtained for the 40 μm film. The authors are suggested to either remove “unusual for films with small thickness (40 micrometer)”, or replace the macroscopic cross-section with the absorption probability. Please also mention the Am-Be neutron source here for easier comparison.

Reply on Comment (6): We agree and have made the necessary revisions in the abstract to reflect this point, as follows.

***Revised manuscript**

(page 1, line 29) Notably, these 2D $\text{Ti}_3\text{C}_2\text{T}_x$ MXene hybrid films exhibited an absorption capacity of 39.8% for neutrons emitted from a ^{241}Am - ^9Be source, unusual for films with small thickness (40 micrometer) and a half-value layer of approximately 54 micrometer thickness, which is among the best values reported for synthetic B_4C composite to date.

7. P.10, second paragraph of the conclusion, the authors state that the commercially available materials possess low B_4C loading, but immediately in the next sentence mention that MirroBor™ has a mass content over 83%, which is much higher than the authors 60 wt.% MBP sample and the best performed MBP sample with 40 wt.%. The two sentences possess contradictory information and should be corrected.

Reply on Comment (7): Thank you for your comment. We have carefully considered your feedback and made the necessary correction in the revised manuscript to resolve the contradictory information. With this revision, the statement is now clear and avoids the contradiction by explaining that commercially available materials generally possess millimeter-scale thickness and, on occasion, exhibit low B_4C loading.

***Revised manuscript**

(page 10, line 26) Commercially available neutron-shielding materials are typically millimeter-scale thick and occasionally exhibit low B loading, as indicated by product specifications. For instance, commercial products such as Mirrobor™ and SWX-238 have thicknesses of 2–5 mm and SWX-238 contains approximately 27.6 wt.% B for a thickness of 3.2 mm.

8. Table S6, please check the effective number of digits. In the authors' description they cut the samples into "20 mm × 20 mm" pieces, i.e. 0.20 cm by 0.20 cm. This does not automatically come to 5 or 6 effective number of digits for the volume and the density.

Reply on Comment (8): In response to the reviewer's comment, we have carefully checked the effective number of digits in Table S6 in the revised manuscript.

***Revised manuscript**

Table S6. Mass densities of MBP hybrid films.

Category	MBP 20	MBP 40	MBP 60
Thickness (μm)	36 ± 2.3	39 ± 2.0	36 ± 2.7
Volume (cm^3)	0.014	0.016	0.014
Weight (g)	0.029	0.030	0.029
Density (g cm^{-3})	2.00	1.95	2.02

Reviewer #2

Remarks to the author: The manuscript has been well revised according to the comment and all the questions have been answered. The manuscript is suggested to be accepted.

Reply: We greatly appreciate the reviewer's insightful and encouraging comments on our work, as well as his/her suggestions to improve the quality of our paper. We thank the reviewer for his/her time and effort.

Reviewers' Comments:

Reviewer #1:

Remarks to the Author:

Appreciations to the authors' efforts on point-to-point clarifications and corrections. There is no further comments to be addressed.

Reviewer #1

Remarks to the author: Appreciations to the authors' efforts on point-to-point clarifications and corrections. There is no further comments to be addressed.

Reply: We greatly appreciate the reviewer's insightful and encouraging comments on our work, as well as his/her suggestions to improve the quality of our paper. We thank the reviewer for his/her time and effort.